# NOVA regulates *Dcc* alternative splicing during neuronal migration and axon guidance in the spinal cord

Janelle C Leggere[1], Yuhki Saito[2], Robert B Darnell[2], Marc Tessier-Lavigne[3], Harald J Junge[1]*, Zhe Chen[1]*

[1]Department of Molecular, Cellular, and Developmental Biology, University of Colorado, Boulder, United States; [2]Laboratory of Molecular Neuro-Oncology, Howard Hughes Medical Institute, The Rockefeller University, New York, United States; [3]Laboratory of Brain Development and Repair, The Rockefeller University, New York, United States

**Abstract** RNA-binding proteins (RBPs) control multiple aspects of post-transcriptional gene regulation and function during various biological processes in the nervous system. To further reveal the functional significance of RBPs during neural development, we carried out an in vivo RNAi screen in the dorsal spinal cord interneurons, including the commissural neurons. We found that the NOVA family of RBPs play a key role in neuronal migration, axon outgrowth, and axon guidance. Interestingly, *Nova* mutants display similar defects as the knockout of the *Dcc* transmembrane receptor. We show here that *Nova* deficiency disrupts the alternative splicing of *Dcc*, and that restoring *Dcc* splicing in *Nova* knockouts is able to rescue the defects. Together, our results demonstrate that the production of DCC splice variants controlled by NOVA has a crucial function during many stages of commissural neuron development.

*For correspondence: harald.junge@colorado.edu (HJJ); zhe.chen@colorado.edu (ZC)

**Competing interests:** The authors declare that no competing interests exist.

## Introduction

Alternative splicing generates gene function complexity in many neural developmental processes, including neuronal differentiation, neuronal migration, axon growth and guidance, and synapse formation and function (*Grabowski and Black, 2001*; *Lee and Irizarry, 2003*; *Li et al., 2007*). A large number of axon guidance molecules undergo alternative splicing, including *Dscam* (*Schmucker et al., 2000*), *Netrin* ligands and their *Dcc* and *Neogenin* receptors (*Keeling et al., 1997*; *Reale et al., 1994*; *Zhang et al., 2004*), *Slit* ligands and *Robo* receptors (*Camurri et al., 2005*; *Chen et al., 2008*; *Clark et al., 2002*; *Dalkic et al., 2006*; *Little et al., 2002*; *Tanno et al., 2004*), *Semaphorin* ligands and *Neuropilin* and *Plexin* receptors (*Cackowski et al., 2004*; *Correa et al., 2001*; *Qu et al., 2002*; *Takahashi et al., 2009*; *Tamagnone et al., 1999*), and *Ephrins* and *Ephs* (*Holmberg et al., 2000*; *Lai et al., 1999*; *Sajjadi et al., 1991*). However, the functional significance of these alternative splicing events and the splicing factors responsible for generating protein variants for these molecules remain largely uncharacterized.

Netrin and *Dcc* (deleted in colorectal carcinoma) function during cell migration, neurite specification and growth, axon guidance, synaptogenesis, and tumorigenesis (*Cooper et al., 1999*; *Killeen, 2009*; *Mehlen and Tauszig-Delamasure, 2014*; *Moore et al., 2007*). Within the spinal cord commissural neurons, DCC is required for Netrin-stimulated axon outgrowth and for attracting the axons to the Netrin-secreting midline (*Dickson, 2002*; *Dickson and Zou, 2010*; *Evans and Bashaw, 2010*). Mammalian *Dcc* undergoes alternative splicing to generate two isoforms that differ in the extracellular domain, in the linker sequence between the fourth and fifth fibronectin repeats (FN4

**eLife digest** The first step of producing a protein involves the DNA of a gene being copied to form a molecule of RNA. This RNA molecule can often be processed to create several different "messenger" RNAs (mRNAs), each of which are used to produce a different protein by a process known as alternative splicing. A class of proteins that bind to RNA molecules controls alternative splicing. These "splicing factors" ensure that the right protein variant is produced at the right time and in the right place to carry out the appropriate activity.

Many genes that play important roles in the nervous system have been reported to undergo alternative splicing to generate different protein variants. However, it is unclear whether alternative splicing is important for controlling how the nervous system develops, during which time the neurons connect to the cells that they will communicate with. Forming these connections involves part of the neuron, called the axon, growing along a precise path through the nervous system to reach its destination. If alternative splicing is important for this process, it is also important to ask: which splicing factors are relevant and which genes do these splicing factors regulate?

Through genetic and molecular studies using mouse embryos, Leggere et al. found that the NOVA family of splicing factors are essential for the development of the nervous system. In particular, the NOVA splicing factors control the alternative splicing of a gene called *Dcc*. This gene produces proteins that play a number of roles, including helping axons to grow and guiding the axons to the correct location in the developing nervous system. A related study by Saito et al. showed that two forms of NOVA splicing factors – called NOVA1 and NOVA2 – have different roles in the nervous system, and describes the role of NOVA2 in more detail.

Leggere et al. will now carry out additional studies to determine the unique role of each protein variant produced from the *Dcc* gene. Future research will also investigate how NOVA proteins help generate these variants at the right time and in the right place.

and FN5) (*Reale et al., 1994*). This alternative splicing was first reported in neuroblastoma cells and was found to be disrupted in these tumor cells (*Reale et al., 1994*), but its physiological significance was completely unknown. We refer to the splice variants hereafter as $DCC_{long}$ and $DCC_{short}$, with $DCC_{long}$ containing extra 20 amino acids in the FN4-FN5 linker. Interestingly, a recent structure study shows that the two isoforms bind Netrin-1 with comparable affinities, but are likely to adopt distinct conformations upon ligand binding (*Xu et al., 2014*).

To reveal the factors that control the alternative splicing of axon guidance genes including *Dcc*, we carried out an in vivo RNAi screen against candidate RNA-binding proteins (RBPs) in cultured mouse embryos, and found that *Nova1* and *Nova2* knockdown leads to severe defects in the dorsal spinal cord interneurons, including the commissural interneurons. The NOVA (neuro-oncological ventral antigen) proteins were first identified as the autoimmune antigens in the neurodegenerative disease POMA (paraneoplastic opsoclonus myoclonus ataxia; *Darnell, 1996*). NOVA1/2 are neural-specific KH (hnRNP K homology)-type of RBPs that can directly regulate alternative splicing (*Buckanovich et al., 1996*; *Lewis et al., 1999*; *Ule et al., 2006*). Genome-wide studies have identified many potential NOVA targets that are involved in various neural developmental processes (*Licatalosi et al., 2008*; *Ule et al., 2003*; *Zhang et al., 2010*). In vivo studies using *Nova* knockout mice have demonstrated defects in synapse formation and function, and in neuronal migration (*Huang et al., 2005*; *Jensen et al., 2000*; *Ruggiu et al., 2009*; *Yano et al., 2010*).

We show here that *Nova1/2* loss of function reduces the migration of the spinal cord interneurons and their progenitors, and disturbs the axon outgrowth and guidance of the commissural interneurons. Interestingly, these defects resemble those seen in *Dcc* knockouts. Consistently, *Dcc* alternative splicing is perturbed by *Nova* deficiency in vivo. Through rescue experiments, we show that restoring $Dcc_{long}$, the diminished isoform in *Nova* knockouts, is able to reverse the defects. Furthermore, NOVA1/2 regulate *Dcc* pre-mRNA splicing in in vitro assays. Together, our results demonstrate that *Dcc* alternative splicing is important for the gene function and is controlled by the NOVA splicing factors.

## Results

### An in vivo RNAi screen to identify RBPs involved in axon guidance

Using the whole mouse embryo culture technique (*Chen et al., 2008*), we transiently knocked down candidate RBPs with small interference RNAs (siRNAs) in the spinal cord and examined the resulting phenotype in commissural axon guidance. We selected RBPs that have neural-specific expression and have been implicated in alternative splicing (*Table 1*). The list of candidates is not exhaustive, and the screening against additional RBPs is ongoing. Some of the RBPs, including NOVA, FOX, and PTBP2/nPTB, directly regulate splicing by recognizing specific sequences in pre-mRNAs and controlling spliceosome assembly (*Agnès and Perron, 2004*; *Li et al., 2007*). Some, such as CELF, UPF, and IGF2BP1, play an indirect role, often by regulating the stability of selective pre-mRNAs and thus influencing their splicing (*Agnès and Perron, 2004*; *Ladd, 2013*; *Li et al., 2007*; *Yap and Makeyev, 2013*). Others, such as ELAVL/HU, can have both direct and indirect effects on alternative splicing (*Agnès and Perron, 2004*; *Li et al., 2007*; *Scheckel et al., 2016*; *Ince-Dunn et al., 2012*).

Using electroporation, we introduced siRNAs and *gfp* into the neural progenitors adjacent to the lumen of the neural tube (*Figure 1A*). We observed that the dorsal interneuron progenitors are mostly targeted, whereas the ventral motor neuron progenitors are not (*Chen et al., 2008*; *Figure 1B*). This is likely due to the fact that the dorsal progenitors are more actively proliferating and differentiating during the culture period. We used *Actb*-gfp (aka *βactin-gfp*) in order to label the highly heterogeneous populations of commissural neurons, which are mostly descended from the dorsal progenitors. Following electroporation, the mitotic neuroprogenitors migrate laterally away from the ventricle (*Caspary and Anderson, 2003*; *Helms and Johnson, 2003*; *Lu et al., 2015*). Upon neural differentiation around E10 in mice, the progenitors exit the cell cycle and migrate out

**Table 1.** RNA-binding proteins targeted in the RNAi screen

| Gene | Protein family | Phenotype | Note |
|------|----------------|-----------|------|
| *Nova1/2* | Neuro-oncological ventral antigen | Neuronal migration, axon outgrowth, and axon guidance defects | |
| *Ptbp2* | Polypyrimidinetract-binding protein | - | |
| *Sfpq* | Splicing factor proline/glutamine rich (polypyrimidine tract-binding protein associated) | - | |
| *Fmr1* | Fragile X mental retardation protein | - | |
| *Nufip1/2* | Nuclear FMRP interacting protein | - | |
| *Elavl1/2/3/4* | ELAV (embryonically lethal abnormal vision) homolog; Hu syndrome protein | Partial midline crossing defect caused by *Elavl2* single RNAi | |
| *Rbfox1/2/3* | RNA-binding protein, Fox (feminizing locus on X) homolog | - | |
| *Msi1* | Musashi RNA-binding protein | - | *Msi1-/-* displays neuronal migration and axon guidance defects in precerebellar neurons but not in spinal commissural neurons (*Kuwako et al., 2010*) |
| *Upf1* | UPF1 regulator of nonsense transcripts homolog | Neuronal migration, axon outgrowth, and axon guidance defects | |
| *Celf1/2/3/4/5/6* | CUG binding protein, Elav-like family member | - | |
| *Khsrp* | KH-type splicing regulatory protein | - | |
| *Pabpc1* | Poly(A) binding protein, cytoplasmic | - | |
| *Igf2bp1* | Insulin-like growth factor 2 mRNA binding protein | Partial midline crossing defect | Regulates *beta-actin* mRNA transport and translation (*Leung et al., 2006*; *Yao et al., 2006*) |
| *Srpk1/2* | Serine/arginine-rich protein specific kinase | - | |

-, no phenotype in spinal cord commissural neurons, when family members were targeted individually.

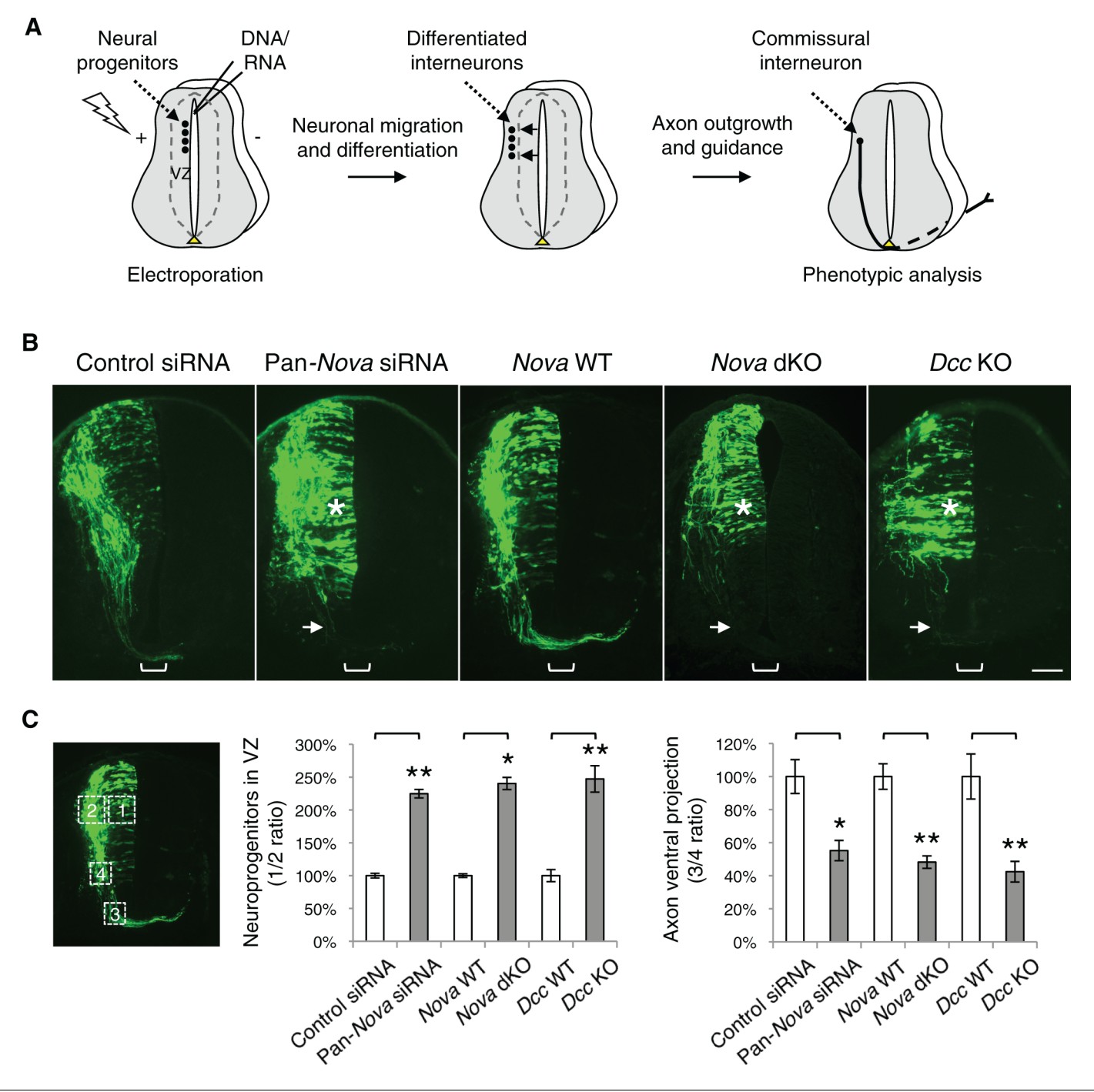

**Figure 1.** *Nova1/2* loss of function disrupts the development of dorsal spinal cord interneurons. (**A**) Schematic of neural development in the dorsal spinal cord during whole embryo culture. Nucleotides are microinjected and electroporated into neural progenitors at the superficial layer of the ventricular zone (VZ, outlined by dashed lines). We observed that the dorsal interneuron progenitors are mostly targeted whereas the ventral motor neuron progenitors are not. The dorsal interneuron progenitors migrate laterally and exit the VZ to differentiate into commissural and ipsilateral interneurons. These interneurons continue to migrate laterally as well as along the dorsoventral axis. Mature commissural neurons extend axons to and across the ventral midline. (**B**) Transverse sections of spinal cords electroporated with control or pan-*Nova* siRNAs, and sections from wildtype (WT), *Nova* double knockout (dKO), and *Dcc* KO. *Actb-gfp* was used to label the highly heterogeneous interneuron populations. *Nova* and *Dcc* deficiency leads to an increased number of neuroprogenitors in the VZ (asterisk), and fewer and shorter ventrally-projecting axons (arrow). Bracket, the ventral midline. (**C**) Quantification of phenotypes in B. Neuroprogenitors in the VZ is quantified as the ratio between the signal from the medial spinal cord (boxed area 1) to that from the lateral spinal cord (area 2). Axon ventral projection is quantified as the ratio between the signal from axons that have

*Figure 1 continued on next page*

*Figure 1 continued*

reached the ventral margin (area 3) and that from the beginning of the axon shaft (area 4). For quantification of phenotypes in all experiments, if *Nova* WT and dKO were not littermates, they were first normalized to the respective double heterozygous (dHet) littermates, and were then compared with each other. *Dcc* KOs were compared with WT littermate controls. Also see *Figure 1—source data 1* for additional quantification. Data are represented as the mean ± SEM (Student's t-test, *p<0.05, **p<0.001). Scale bar, 50 μm.

The following source data and figure supplements are available for figure 1:

**Source data 1.** Quantification of neuronal migration and axon projection phenotypes in cultured embryos.
**Figure supplement 1.** RNAi phenotypes of candidate RNA-binding proteins.
**Figure supplement 2.** *Nova* and *Dcc* KOs cause axon projection defect.

of the ventricular zone (VZ) to reach the lateral spinal cord. Differentiated interneurons, including commissural and ipsilateral-projecting neurons continue to migrate laterally and different subpopulations also migrate dorsally or ventrally to reach their final positions. Mature commissural neurons then grow out axons and project them toward and across the ventral midline (*Caspary and Anderson, 2003*; *Helms and Johnson, 2003*; *Lu et al., 2015*; *Figure 1A*).

Among the candidates, knockdown of *Nova1/2*, *Upf1*, *Elavl2*, and *Igf2bp1* caused commissural neuron defects (*Figure 1—figure supplement 1*). *Nova1/2* and *Upf1* knockdown disrupted both neuronal migration and axon projection (*Figure 1—figure supplement 1A* and see below). *Elavl2* and *Igf2bp1* knockdown partially blocked commissural axons from crossing the midline (*Figure 1—figure supplement 1B*). Since many RBPs have multiple family members, the lack of effect from other candidates may result from functional redundancy between homologues.

## *Nova1/2* loss of function disrupts dorsal interneuron development and resembles *Dcc* knockout

We found that single knockdown of either *Nova1* or *Nova2* had no effect compared with control siRNAs, but a pan-*Nova* siRNA that targets both homologues caused severe defects (*Figure 1B,C*). First, there were an increased number of GFP-positive neurons within the VZ, indicating an abnormality in the interneuron progenitors. Second, there were fewer ventrally projecting axons, which often failed to reach the midline, suggesting a defect in commissural axon growth and/or guidance (*Figure 1B,C*, *Figure 1—source data 1*).

To corroborate the RNAi effect, we examined *Nova1/2* knockout (KO) embryos labeled with *Actb-gfp*. We found that neither *Nova1* nor *Nova2* single KOs displayed any defect. However, *Nova1* KO; *Nova2* KO (referred to hereafter as *Nova* dKO) caused the same defects as *Nova1/2* double knockdown (*Figure 1B,C*, *Figure 1—source data 1*). In addition, *Nova1* Het; *Nova2* KO displayed the same defects, whereas other genotype combinations including *Nova1* KO; *Nova2* Het are phenotypically normal (see below). Thus, *Nova1* and *Nova2* have redundant functions in regulating dorsal interneuron development and *Nova2* appears to play a major role.

The Netrin-DCC signaling is required to stimulate commissural axon outgrowth and to attract the axons to the ventral midline, we thus compared cultured *Dcc* KOs with *Nova* dKOs. Interestingly, we observed the same defects in *Dcc* KOs, namely more GFP+ neurons within the VZ, and shorter and fewer ventrally projecting axons (*Figure 1B,C*, *Figure 1—source data 1*). We also labeled *Nova* and *Dcc* KOs with *Atoh1-gfp* (aka *Math1-gfp*), which is expressed in the dorsal most dI1 subpopulation of interneurons and their progenitors (*Lumpkin et al., 2003*). Similarly, we observed a reduction in axon ventral projection in both mutants (*Figure 1—figure supplement 2*, *Figure 1—source data 1*). Due to the small number of neurons being labeled, we could not determine if there are more *Atoh1-gfp* expressing neurons in the VZ in the mutants.

To determine the identity of the ACTB-GFP+ neurons and axons, we immunostained the cultured embryos with neuronal and axonal markers. We used PAX3/7 antibodies to label the dorsal interneuron progenitors that give rise to different populations of commissural and ipsilateral neurons (*Caspary and Anderson, 2003*; *Helms and Johnson, 2003*). *Pax3* and *Pax7* are required to specify the majority of commissural neurons, as their double knockout greatly reduces the ventral

commissure formed by commissural axons crossing the midline (*Mansouri and Gruss, 1998*). We labeled differentiated interneurons with antibodies against BARHL2 and LHX5, two transcription factors among many others that are expressed by post-mitotic interneurons. Combinations of these transcription factors define the different lineages of commissural and ipsilateral neurons (*Caspary and Anderson, 2003*; *Helms and Johnson, 2003*). We also labeled commissural axons specifically with anti-ROBO3 (*Sabatier et al., 2004*). Our studies show that the GFP+ neurons within the VZ are dorsal interneuron progenitors and the ones at the lateral spinal cord are differentiated interneurons. In addition, the GFP+ axons that fail to project to the midline are commissural axons (*Figure 2*). Furthermore, using fluorescent in situ hybridization, we found that the GFP+ neuroprogenitors and interneurons also express *Dcc* (*Figure 2—figure supplement 1*), consistent with previous reports that *Dcc* is expressed in these neuronal populations (*Keino-Masu et al., 1996*; *Phan et al., 2011*). Taken together, *Nova* deficiency appears to disturb commissural axon projection and also interferes with earlier stages of commissural neuron development in the progenitors.

### *Nova* dKO delays neuronal migration but does not affect neuronal differentiation

The increase in GFP+ neuroprogenitors in the VZ could result from slowed migration, increased proliferation, or reduced neuronal differentiation. To follow cell migration, we electroporated the progenitors with *Actb-gfp* and cultured the embryos for 20 hrs. In WT embryos, we observed that many progenitors have reached the lateral spinal cord after 20 hrs. However, in both *Nova* and *Dcc* KOs, almost all GFP+ neurons are still positioned within the VZ (*Figure 3*). This supports the idea that neuronal migration is reduced. We examined cell proliferation in *Nova* and *Dcc* KOs using cell cycle markers, including phospho-histone H3 (pH3), a mitotic marker, and Ki-67, a cell proliferation marker. We found that at E10.5, the number of neural stem cells and progenitors is normal in both mutants (*Figure 3—figure supplement 1*). In addition, we labeled the neural progenitors in the whole spinal cord with SOX2 and the dorsal progenitors with PAX3/7, and found no change in the localization or organization of these progenitor populations (*Figure 3—figure supplement 1*). We also examined interneuron differentiation using the BARHL2, ISL1/2, and LHX5 markers, and found that a normal number of interneurons are born in *Nova* and *Dcc* KOs at E10.5 (*Figure 3—figure supplement 2*; *Figure 4*). Therefore, both *Nova* and *Dcc* deficiency reduces neuroprogenitor migration, but does not affect cell proliferation or interneuron differentiation.

We further examined the migration of post-mitotic interneurons using the BARHL2 marker. *Barhl2* acts downstream of *Atoh1* to specify the dI1 interneurons, including the dI1i (ipsilateral) and dI1c (commissural) populations (*Ding et al., 2012*; *Figure 4A*). DI1 neurons migrate both ventrally and laterally, and dI1i migrates even further laterally than dI1c (*Ding et al., 2012*). As discussed above, a normal number of BARHL2+ neurons are generated at E10.5 in *Nova* and *Dcc* KOs (*Figure 4*). By E11.75, most neurons in WT have migrated ventrally and some start to arrive at the lateral margin of the spinal cord (*Ding et al., 2012*; *Figure 4*). In contrast, there are fewer BARHL2+ neurons at the ventral most and lateral most positions in *Nova* and *Dcc* KOs (*Figure 4*), suggesting a reduction in both dorsoventral and mediolateral migration. Later at E12.5, the dI1c and dI1i populations can be well discerned in WT (*Ding et al., 2012*; *Figure 4A*). Similarly, *Nova* and *Dcc* KOs also had two distinct populations of BARHL2+ neurons and the ratio between them is normal (*Figure 4*). Therefore, despite a transient delay in the migration, both commissural and ipsilateral neurons are properly specified.

### *Nova* dKOs disrupts Netrin-DCC dependent commissural axon outgrowth

To understand the defect in commissural axon ventral projection, we examined axon outgrowth in *Nova* dKO using explants of the dorsal spinal cord (DSC). In this assay, commissural axons extend out of the explant in the presence of Netrin-1 after a culture period of 16 hrs (*Serafini et al., 1994*). In the absence of Netrin-1, commissural axons are also able to grow out, but only after an extended period of culturing (after 40 hrs; *Keino-Masu et al., 1996*). We isolated E11.5 DSC from *Nova* dKOs and first tested if commissural neurons are defective in Netrin-stimulated axon outgrowth. Even after 24 hrs in the presence of Netrin-1, when control explants had robust axon outgrowth, *Nova* dKO

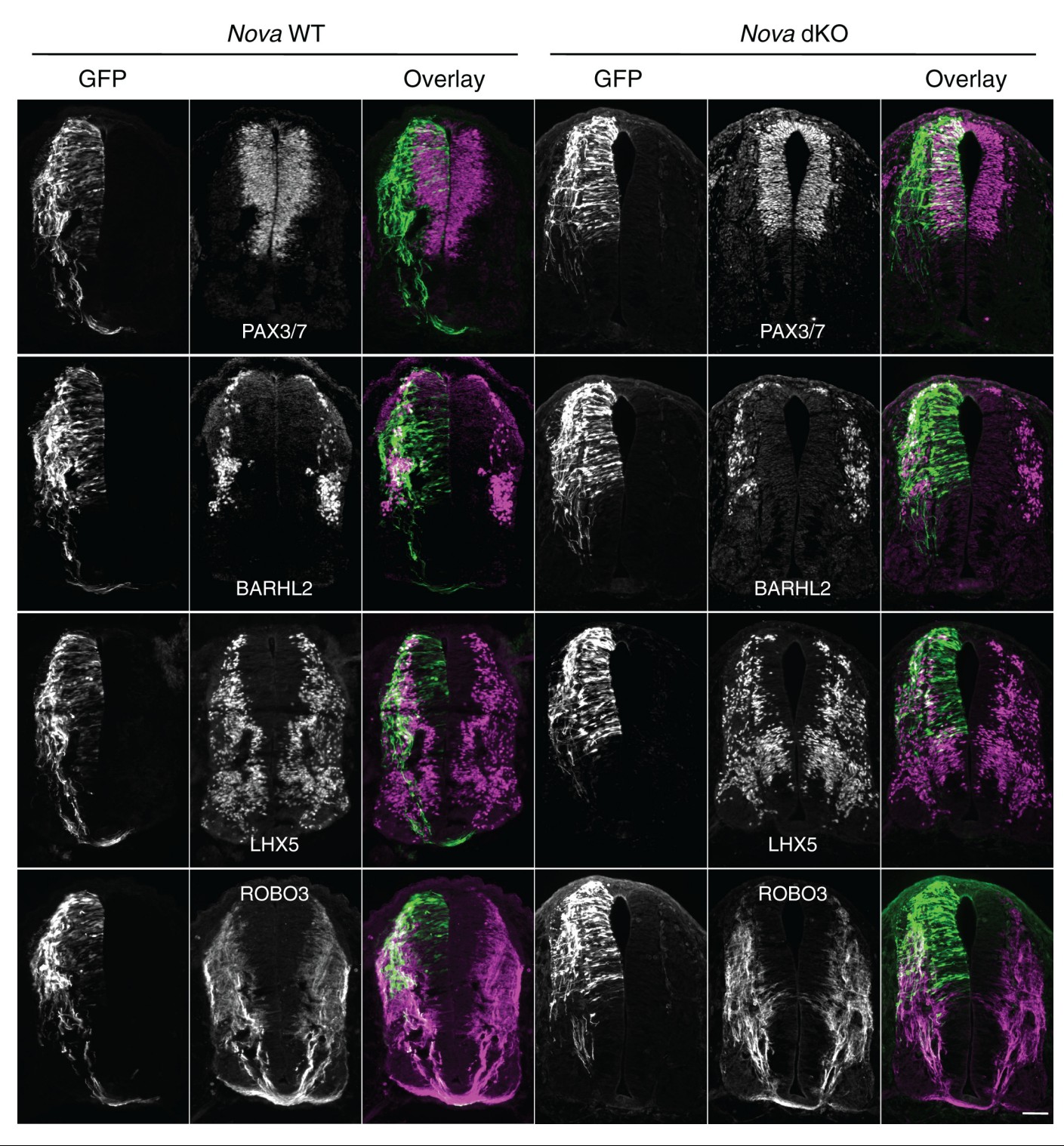

**Figure 2.** Expression of neuronal and axonal markers in cultured *Nova* WT and dKO embryos electroporated with *Actb-gfp*. PAX3/7 immunostaining labels dorsal interneuron progenitors, which reside within the ventricular zone and give rise to different populations of interneurons. BARHL2 and LHX5 stainings label differentiated interneurons at the lateral spinal cord. ROBO3 staining specifically labels the commissural axons. Scale bar, 50 μm.

The following figure supplement is available for figure 2:

**Figure supplement 1.** Fluorescent in situ hybridization of *Dcc* in cultured embryos electroporated with *Actb-gfp*.

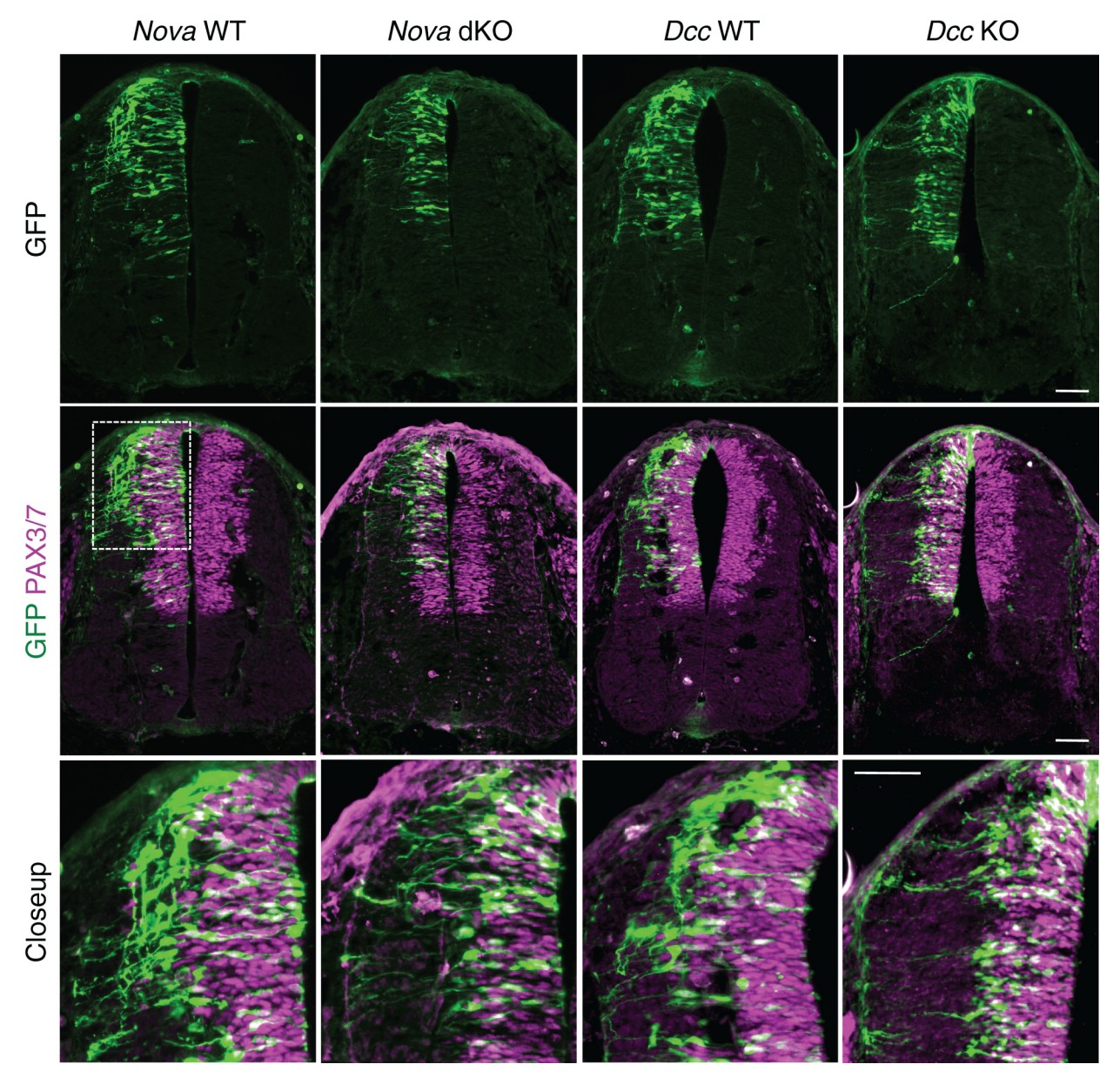

**Figure 3.** *Nova* and *Dcc* knockouts delay the lateral migration of dorsal interneuron progenitors. *Actb-gfp* was electroporated into neural progenitors and embryos were cultured for 20 hrs. The bottom panel shows the closeup images of the boxed area from the middle panel. Many WT neurons have migrated out of the VZ (demarcated by PAX3/7 staining) after 20 hrs, whereas *Nova* and *Dcc* KO neurons are mostly located within the VZ. Scale bars, 50 μm.

The following figure supplements are available for figure 3:

**Figure supplement 1.** Neural stem cells and progenitors appear normal in *Nova* and *Dcc* KOs.

**Figure supplement 2.** Dorsal interneuron differentiation is normal in *Nova* and *Dcc* KOs.

axons had very little growth (*Figure 5*). The same outgrowth defect was also seen in *Dcc* KOs, as previously reported (*Xu et al., 2014*; *Figure 5*).

We also examined Netrin-independent axon outgrowth by culturing the explants in the absence of Netrin-1 for 40 hrs (*Figure 5*). As expected, we did not observe any axon outgrowth after 24 hrs.

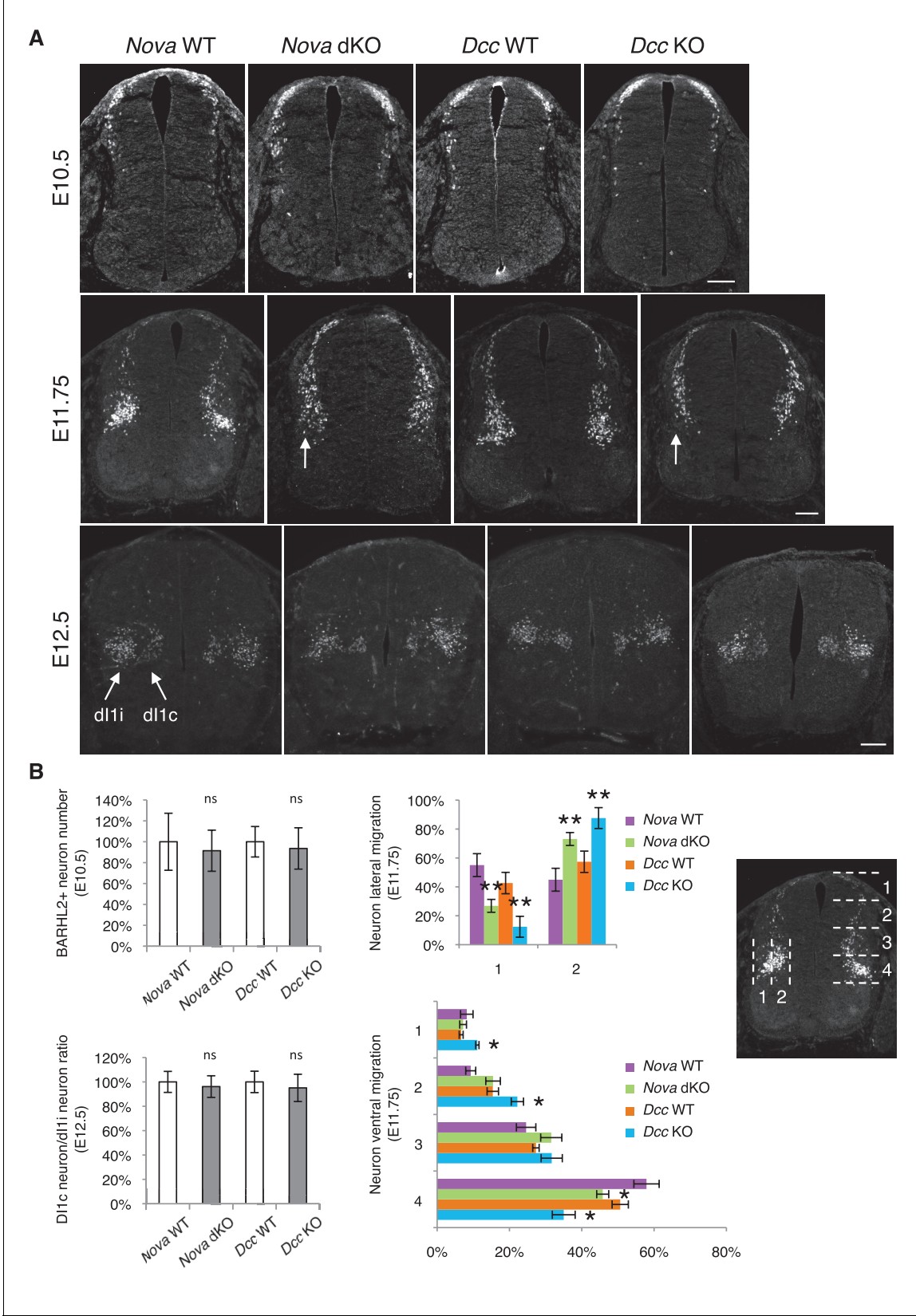

**Figure 4.** *Nova* and *Dcc* knockouts delay neuronal migration but do not disturb neuronal differentiation. (**A**) Immunohistochemistry of BARHL2 in *Nova* WT, *Nova* dKO, *Dcc* WT, and *Dcc* KO spinal cords. A normal number of BARHL2+ neurons are generated in *Nova* and *Dcc* mutants at E10.5. At E11.75, *Figure 4 continued on next page*

*Figure 4 continued*

BARHL2+ neurons fail to arrive at the lateral margin of the spinal cord in the KOs (indicated by arrow). Their ventral migration is also delayed. At E12.5, both dI1i (ipsilateral) and dI1c (commissural) populations appear normal in the mutants. (**B**) Quantification of phenotypes in A. For quantification of the lateral migration, the distance between the lateral margin of the spinal cord and the medial most BARHL2+ neurons are divided into two. The percentage of neurons within each half is shown. For quantification of the ventral migration, the distance from the dorsal margin of the spinal cord to the ventral most BARHL2+ neurons are divided into four. The percentage of neurons within each quarter (1 to 4 from dorsal to ventral) is shown. Three embryos from each genotype and at least five sections from each embryo were quantified. Data are normalized to WT for total neuron number and dI1c/dI1i ratio. Data are represented as the mean ± SEM (Student's t-test for total neuron number, dI1c/dI1i ratio, and lateral migration, **p<0.001, ns, not significant. Two way ANOVA and Bonferroni post test for ventral migration, *p<0.05). Scale bars, 50 μm.

However after 40 hrs, there was robust axon outgrowth from both control and *Nova* dKO, and the degree of growth is comparable between the two. In addition, *Dcc* KOs also displayed the same degree of Netrin-independent axon outgrowth (*Figure 5*). Therefore, like *Dcc* KOs, *Nova* dKO commissural axons do not have a general growth defect. Rather, they fail to grow in response to Netrin stimulation.

## *Nova* dKOs disrupts commissural axon guidance to the midline

Loss of Netrin-DCC mediated attraction in vivo reduces the number of commissural axons that are able to reach the midline (*Fazeli et al., 1997*; *Serafini et al., 1996*; *Xu et al., 2014*). We thus examined commissural axon guidance to the midline using immunostaining of the axonal markers ROBO3 and TAG-1. We found that at E10.5 and E11.5, the intensity of these axonal markers from the ventral half of the spinal cord was significantly reduced in *Nova* dKOs, suggesting a reduction of ventral axon projection (*Figure 6*). At E11.5, there are usually two main commissural axon bundles. Both are reduced in *Nova* dKOs and the more lateral one is more profoundly affected. The size of the ventral commissure, formed by axons crossing the midline, is also reduced (*Figure 6*). By E12.5, the reduction in the commissure size is still significant but is somewhat alleviated than at earlier stages. A similar reduction in axons reaching and crossing the ventral midline is also seen in *Dcc* KOs (*Fazeli et al., 1997*; *Xu et al., 2014*). The severity of the defect is similar between *Nova* and *Dcc* mutants (*Figure 6*). One distinction between the two mutants is that *Dcc* KO axons are defasciculated and often invade the motor column (*Xu et al., 2014*), whereas such a defect was not observed in *Nova* dKOs (*Figure 6—figure supplement 1*).

Besides Netrin, SHH (sonic hedgehog) and VEGF (vascular endothelial growth factor) have also been shown to attract commissural axons (*Charron et al., 2003*; *Ruiz de Almodovar et al., 2011*). However, the loss of SHH or VEGF and their receptors in vivo does not lead to a significant reduction of the ventral commissure (*Charron et al., 2003*; *Okada et al., 2006*; *Ruiz de Almodovar et al., 2011*). SHH and VEGF also do not induce commissural axon outgrowth in the DSC assay, and their deficiency does not affect commissural axon outgrowth in vivo. Taken together, our results show that *Nova* loss of function disrupts Netrin-DCC signaling during commissural axon outgrowth and guidance.

## *Nova* dKO disrupts *Dcc* alternative splicing

To determine if the Netrin-DCC pathway is affected on the molecular level, we set out to examine the expression and alternative splicing of the pathway components. Using in situ hybridization, we found that *Nova1* and *Nova2* are highly expressed in commissural neurons, but not in the floorplate, where Netrin is expressed (*Figure 7—figure supplement 1*). This expression pattern as well as the fact that isolated *Nova* dKO neurons fail to extend axons in response to Netrin suggest that *Nova1/2* most likely function cell autonomously within commissural neurons. Using quantitative RT-PCR and western blotting, we first examined the total mRNA and protein levels of *Dcc*. We found that they were not significantly altered in *Nova* dKOs, although there was a slight increase in both levels (*Figure 7—figure supplement 2A,B*).

Both human and mouse *Dcc* undergoes alternative splicing at exon 17 to give rise to $Dcc_{long}$ and $Dcc_{short}$, with $Dcc_{long}$ containing extra 60 bp (*Reale et al., 1994*; *Figure 7A*). Human *Dcc* undergoes additional alternative splicing in multiple regions in tumor cells (*Reale et al., 1994*; *Huerta et al., 2001*). By RT-PCR using spinal cord tissues from both wildtype and *Nova* dKO, we found that mouse

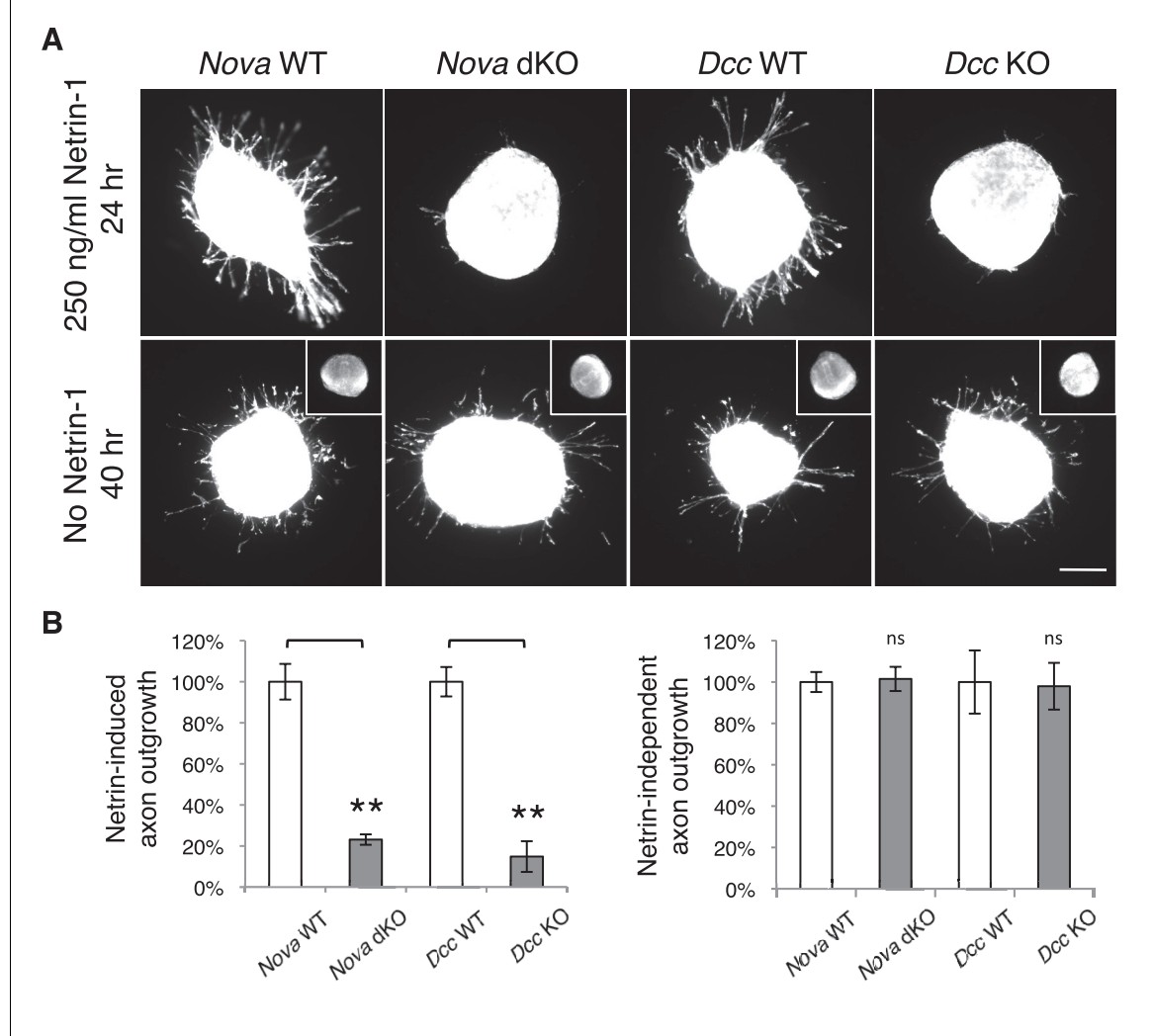

**Figure 5.** *Nova* dKO disrupts Netrin-1 induced axon outgrowth in dorsal spinal cord (DSC) explant assays. (**A**) DSC axon outgrowth in *Nova* WT, *Nova* dKO, *Dcc* WT, and *Dcc* KO in the presence or absence of 250 ng/ml Netrin-1. Axons were visualized by rhodamine-phalloidin staining. *Nova* dKO and *Dcc* KO display a drastic reduction in axon growth in response to Netrin. However, both mutants are able to extend axons after an extended culture period in the absence of Netrin. The insets in the bottom panel show no axon outgrowth after 24 hrs of culturing for all genotypes in the absence of Netrin. (**B**) Quantification of axon outgrowth in A. Axon outgrowth is represented as the ratio between the signal from all axons extending out of the explant and that from the cell bodies within the explant, after background extraction. Three embryos from each genotype and at least five explants from each embryo were quantified. Data are normalized to WT and are represented as the mean ± SEM (Student's t-test, **$p < 0.001$, ns, not significant). Scale bar, 50 μm.

The following figure supplement is available for figure 5:

**Figure supplement 1.** Commissural axon outgrowth in *Nova* embryos.

*Dcc* does not produce alterative mRNAs in these additional regions. To determine if there are previously unknown alternative splicing events in mouse *Dcc*, we amplified and sequenced overlapping cDNA fragments from CD-1 mice, and did not identify any additional alternative splicing.

Using quantitative RT-PCR, we specifically examined *Dcc* alternative splicing at exon 17 and found that it was significantly disturbed in *Nova* dKOs. $Dcc_{long}$ was greatly diminished, whereas $Dcc_{short}$ was upregulated (*Figure 7B,C*). In addition, the changes appear to be sensitive to *Nova1/2* gene copy number. When *Nova2* is reduced from two copies (wildtype) to one (heterozygote) and further to zero (knockout), $Dcc_{long}$ decreases and $Dcc_{short}$ increases stepwise (*Figure 7B*). *Nova1* KO;

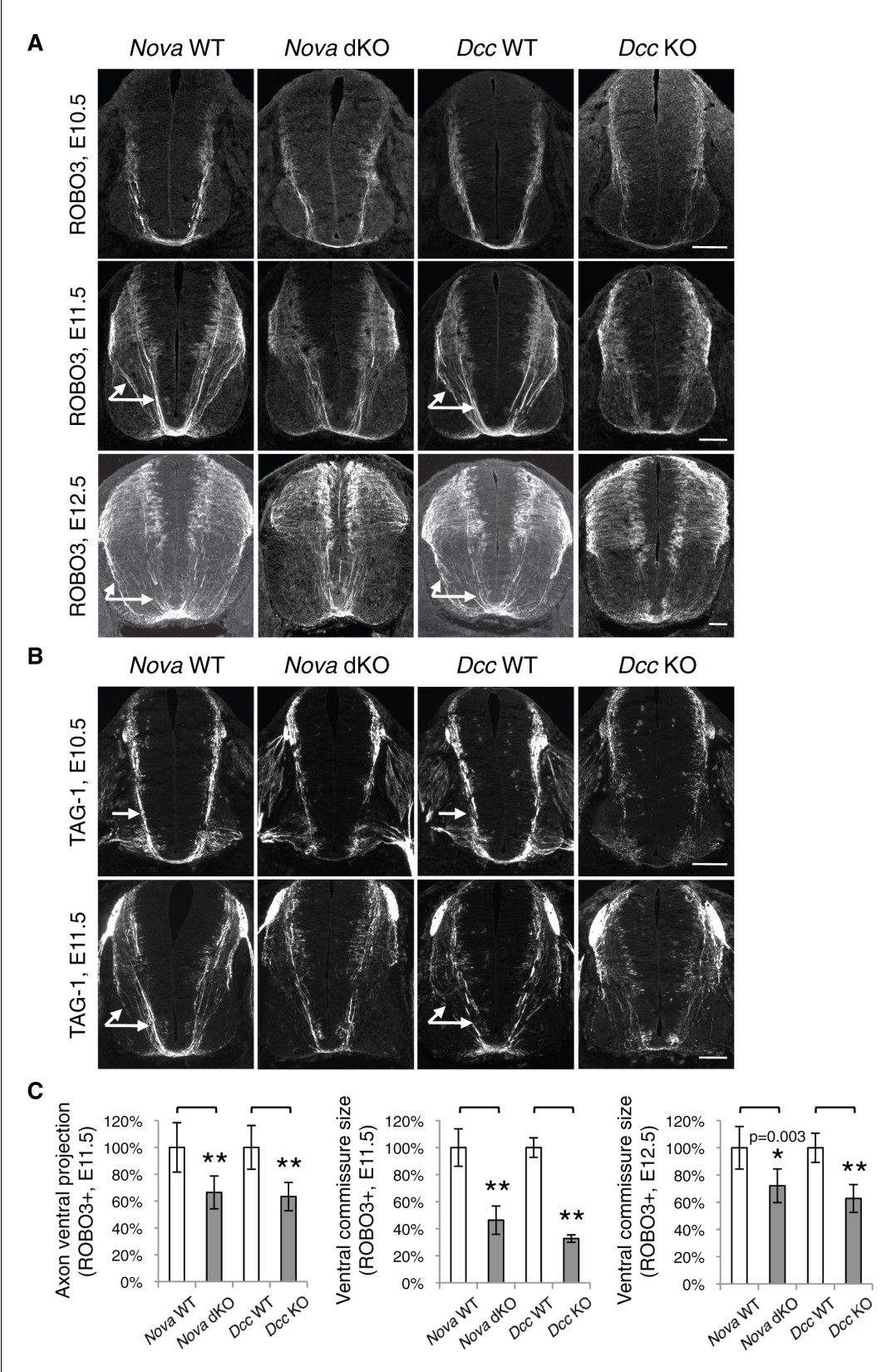

**Figure 6.** *Nova* dKO disrupts commissural axon ventral projection. (**A**) and (**B**) Immunohistochemistry of commissural axonal markers, ROBO3 and TAG-1 respectively, in transverse sections of the spinal cord. Two main axon bundles can be observed at E11.5 (arrows) and the more lateral one is severely

*Figure 6 continued on next page*

*Figure 6 continued*

reduced in *Nova* dKOs. In the top panel of B, arrows indicate commissural axon projection, and the staining in the ventral lateral spinal cord is from motor axons. (C) Quantification of axon guidance phenotypes in A. To quantify axon ventral projection, the signal from the ventral half of the spinal cord was first normalized to the signal from the whole spinal cord, and then normalized to WT controls. The commissure size is represented as the ratio between the thickness of the axon bundle at the midline and that of the floorplate. Data are normalized to WT. Three embryos from each genotype and at least five sections from each embryo were quantified. Data are represented as the mean ± SEM (Student's t-test, *p<0.05, **p<0.001). Scale bars, 50 μm.

The following figure supplement is available for figure 6:

**Figure supplement 1.** *Nova* dKO does not cause commissural axons to invade the motor column.

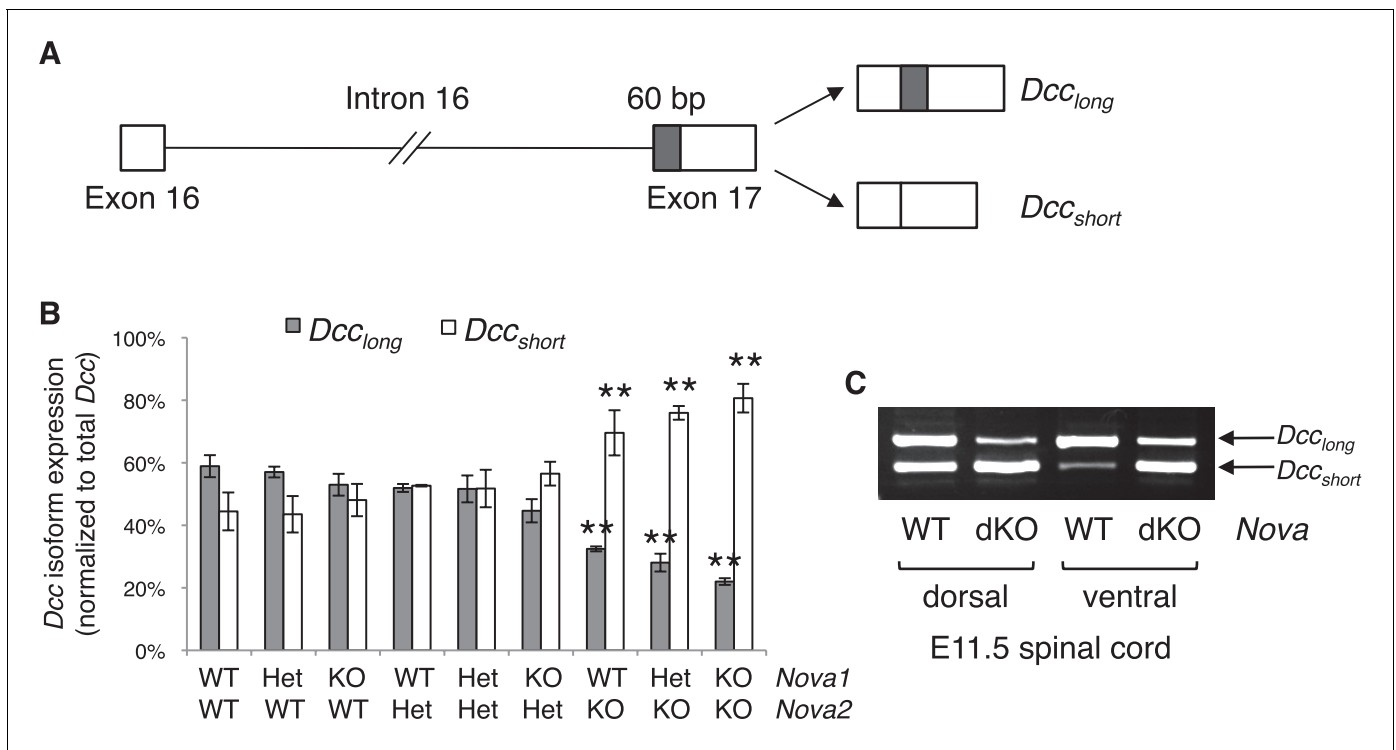

**Figure 7.** *Nova* dKO disrupts the alternative splicing of *Dcc*. (A) Schematic of *Dcc* alternative splicing and the resulting isoforms. The alternative 60 bp in exon 17 is shaded grey. (B) *Dcc* isoform expression in nine *Nova1/2* genotypes, as measured by quantitative RT-PCR using E11.5 dorsal spinal cord. The isoform expression level is normalized to a common region in *Dcc* cDNA (total *Dcc*; see materials and methods). Note that although the changes in *Nova2* single KOs are statistically significant, the in vivo phenotype of *Nova2* KOs is not significantly different from controls (see **Figure 5—figure supplement 1**). (C) *Dcc* isoform expression from *Nova* embryos detected by semi-quantitative PCR that amplify both isoforms. Dorsal and ventral spinal cords are separated to distinguish commissural neurons (dorsally located) and motor neurons (ventrally located). *Nova* dKO affects *Dcc* alternative splicing in both populations. Three animals from each genotype were quantified. Data are represented as the mean ± SEM (one way ANOVA and Bonferroni post test, **p<0.001).

The following figure supplements are available for figure 7:

**Figure supplement 1.** *Nova1* and *Nova2* expression in E11.5 spinal cord detected by in situ hybridization.

**Figure supplement 2.** *Nova* dKO does not affect the total level of *Dcc, Neo1*, and *Robo3* expression.

**Figure supplement 3.** *Nova* dKO does not affect the alternative splicing of *Neo1, Robo3,* or *Epha5*.

*Nova2* KO has the most significant changes among all genotypes. Consistently, the defects seen in *Nova1* KO; *Nova2* KO is the most severe (*Figure 5—figure supplement 1*). Together, these results suggest that *Nova* normally regulates *Dcc* alternative splicing by promoting $Dcc_{long}$ in a dose-sensitive manner. It is interesting to note that even though the changes in *Nova2* single KOs are statistically significant, no defect was seen in this genotype (*Figure 5—figure supplement 1*). This suggests that there may be a threshold for the level of *Dcc* isoforms for normal function. Alternatively, the somewhat elevated total *Dcc* expression (*Figure 7—figure supplement 2A,B*) may be able to compensate for a small disruption of activity. Consistent with our results, accompanying paper by Saito et al. also found that the alternative splicing at *Dcc* exon 17 is significantly altered in E18.5 *Nova2* KO cortex, which may contribute to additional axon guidance defects in the brain.

Several molecules have been suggested to function as Netrin co-receptors within commissural neurons, including Neogenin (NEO1), ROBO3, DSCAM, and APP (*Ly et al., 2008*; *Rama et al., 2012*; *Xu et al., 2014*; *Zelina et al., 2014*). Loss of function in *Neo1*, a *Dcc* homolog and a NOVA target (see accompanying paper by Saito et al.), does not affect commissural axon outgrowth or guidance by itself, but can enhance the defects of *Dcc* KOs (*Xu et al., 2014*). The ROBO3 receptor, specifically the $ROBO3_{A.1}$ isoform, represses premature repulsion before axons reach the midline and also forms a complex with DCC to mediate Netrin attraction (*Chen et al., 2008*; *Sabatier et al., 2004*; *Zelina et al., 2014*). DSCAM and APP have not been shown to be required for Netrin activity in vivo (*Palmesino et al., 2012*; *Rama et al., 2012*).

Using primers that detect different regions of the cDNAs, we found that the total mRNA level of *Neo1* and *Robo3* was not affected (*Figure 7—figure supplement 2C,D*). We also examined *Neo1* alternative splicing in a homologous region to *Dcc* exon 17, and found that it was not significantly altered in the dorsal spinal cord in *Nova* dKOs (*Figure 7—figure supplement 3A*). Other alternative regions of *Neo1* (*Keeling et al., 1997*) were not significantly altered either (*Figure 7—figure supplement 3A*). *Robo3* pre-mRNA undergoes alternative splicing in multiple regions (*Camurri et al., 2005*; *Chen et al., 2008*; *Yuan et al., 1999*). We measured the expression levels of these alternative areas, as well as the adjacent constant exons, and found no change in any of them in *Nova* dKOs compared with controls (*Figure 7—figure supplement 3B*). For further comparison, we also examined the alternative splicing of the *Epha5* receptor, which is a known NOVA target but has not been shown to play a role in the development of the dorsal spinal cord interneurons. We did not observe any change in the alternative splicing of *Epha5* exon 7 (*Figure 7—figure supplement 3C*), which is found to be altered in E18.5 *Nova2* KO cortex (see accompanying paper by Saito et al.).

Together, these results show that *Dcc* is the most likely target of *Nova1/2* that is affected within commissural neurons, while other receptors are unlikely to be responsible for the *Nova* dKO defects.

## *Nova* dKOs defects can be rescued by restoring $Dcc_{long}$ expression, but not by $Dcc_{short}$ expression

If the *Nova* dKO defects are indeed caused by disturbed *Dcc* alternative splicing, with $Dcc_{long}$ greatly reduced, restoring $Dcc_{long}$ in the mutants should be able to reverse the defects. We first used the axon outgrowth assay to compare the rescuing abilities of *Dcc* isoforms. We electroporated either isoform into the spinal cords at E10.5, cultured the embryos for one day to allow exogenous protein expression, and then carried out the DSC assay. As shown above, *Nova* dKO axons fail to grow in response to Netrin-1 (*Figure 5*; *Figure 8A,B*). When $Dcc_{long}$ was introduced back into *Nova* dKOs, many axons extended out of the explant, whereas $Dcc_{short}$ was unable to rescue the defect (*Figure 8A,B*). For comparison, we also overexpressed $Robo3_{A.1}$ in *Nova* dKOs, and found that it was unable to restore commissural axon outgrowth in *Nova* dKOs (*Figure 8A,B*).

We then introduced *Dcc* isoforms back into cultured embryos to determine if restoring $Dcc_{long}$ could rescue additional defects seen in *Nova* dKOs. We electroporated the cDNAs and *Actb-gfp* into embryos at E9.5 and cultured the embryos for two days. We found that $Dcc_{long}$ expression in *Nova* dKOs could indeed ameliorate the neuronal migration and axon ventral projection defects (*Figure 8C,D*, *Figure 1—source data 1*). In contrast, overexpression of $Dcc_{short}$, the isoform that is abnormally upregulated, did not rescue. In fact, it exacerbated the defects, causing even fewer axons to project ventrally toward the midline (*Figure 8C,D*, *Figure 1—source data 1*). We also overexpressed $Robo3_{A.1}$ in *Nova* dKOs and found that it did not have any effect on neuronal migration or axon projection (*Figure 8C,D*, *Figure 1—source data 1*).

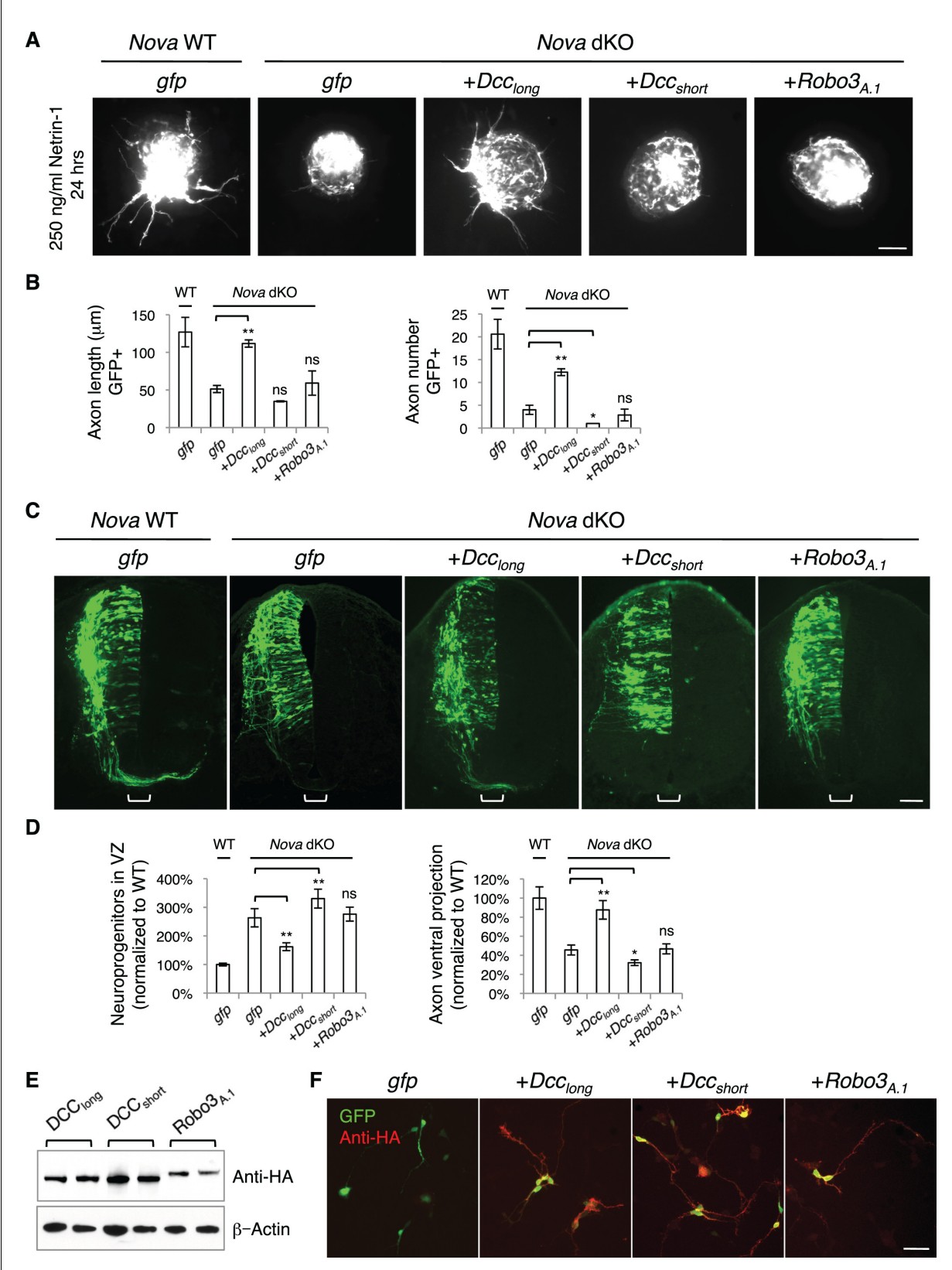

**Figure 8.** Expression of DCC$_{long}$, but not DCC$_{short}$ or Robo3$_{A.1}$, is able to rescue *Nova* dKO defects. (**A**) Dorsal spinal cord assays using *Nova* WT and dKO neurons electroporated with *gfp*, *Dcc$_{long}$*, *Dcc$_{short}$*, or *Robo3$_{A.1}$*. The embryos were electroporated at E10.5 and cultured for one day to allow

*Figure 8 continued on next page*

*Figure 8 continued*

protein expression. DSC assays were then carried out and axons were directly visualized by GFP fluorescence. Explants were cultured for 24 hrs with 250 ng/ml Netrin-1. Only DCC$_{long}$ expression can rescue the outgrowth defect in *Nova* dKOs. (B) Quantification of DSC axon length and number in A. Three embryos from each treatment/genotype and at least five explants from each embryo were quantified. (C) Transverse sections of spinal cords from *Nova* WT and dKO electroporated with *gfp*, *Dcc$_{long}$*, *Dcc$_{short}$*, or *Robo3$_{A.1}$*. Only *Dcc$_{long}$* is able to rescue the neuronal migration and axon projection defects. Bracket, ventral midline. (D) Quantification of phenotypes in C (see description in *Figure 1C*). Also see *Figure 1—source data 1* for additional quantification. (E) Expression of DCC$_{long}$, DCC$_{short}$, and ROBO3$_{A.1}$ proteins in electroporated embryos (two embryos are shown). All three proteins were immunoprecipitated by a C-terminal HA tag and detected by western blotting. (F) Expression of DCC$_{long}$, DCC$_{short}$, and ROBO3$_{A.1}$ proteins in dissociated dorsal spinal cord neurons, as detected by anti-HA. Data are represented as the mean ± SEM (one way ANOVA and Bonferroni post test, *p<0.05, **p<0.001, ns, not significant). Scale bars, 50 µm.

The following figure supplement is available for figure 8:

**Figure supplement 1.** Rescue of *Nova* dKOs by *Dcc* isoforms in embryos labeled with *Atoh1-gfp*.

To confirm the expression of the exogenous proteins, which are tagged with an HA peptide at the very C-terminus, we performed immunoprecipitation from electroporated embryos and analyzed the protein levels by SDS-PAGE and western blotting. All proteins were expressed at detectable and comparable levels (*Figure 8E*). We also dissociated electroporated spinal cords and examined the protein expression within GFP+ neurons. Using HA antibodies, we were able to detect all three proteins within axons (*Figure 8F*), showing that the proteins are properly expressed and localized. Therefore, the rescue of *Nova* dKOs only by *Dcc$_{long}$* suggests that the defects result directly from a loss in *Dcc* activity, in particular *Dcc$_{long}$* activity. The inability of ROBO3$_{A.1}$ to rescue *Nova* dKOs is consistent with the fact that *Robo3* expression and alternative splicing are not disrupted by *Nova* dKO.

Furthermore, using the *Atoh1-gfp* marker, we also found that *Dcc$_{long}$* expression, but not *Dcc$_{short}$* expression was able to rescue the axon projection defect in *Nova* dKOs to a large extent (*Figure 8— figure supplement 1*, *Figure 1—source data 1*).

## NOVA1/2 regulate the alternative splicing of *Dcc* pre-mRNA

NOVA proteins specifically recognize clusters of YCAY (Y=C/U) sequences (*Buckanovich and Darnell, 1997*). A genome-wide study using HITS-CLIP (high-throughput sequencing of RNA isolated by crosslinking immunoprecipitation) has identified candidate target sites within *Dcc* pre-mRNA (*Zhang et al., 2010*). To determine if NOVA1/2 directly regulate *Dcc* alternative splicing, we carried out splicing assays using a *Dcc* minigene containing the genomic DNA between exons 16 and 17 (*Figure 9A*). The alternative sequence is located within exon 17 and the candidate YCAY clusters are located within exon 16, intron 16, and exon 17. We coexpressed the minigene with an empty vector or *Nova1*, *Nova2*, or *Ptbp2* (an unrelated splicing factor that has not been shown to affect *Dcc* alternative splicing). When the vector alone was coexpressed, we detected two RT-PCR products corresponding to *Dcc$_{long}$* and *Dcc$_{short}$*, respectively (*Figure 9B*). When *Nova1* or *Nova2* was coexpressed, *Dcc$_{long}$* was upregulated, while *Dcc$_{short}$* was reduced. Such changes were not observed when *Ptbp2* was coexpressed (*Figure 9B*). These results suggest that NOVA1/2 promote *Dcc$_{long}$*, consistent with the observed reduction of *Dcc$_{long}$* in *Nova* dKOs. Therefore, the in vitro splicing assays recapitulate *Dcc* alternative splicing pattern in vivo.

To further determine the specificity of the effect of NOVA, we mutated the YCAY repeats to YAAY, which can no longer be recognized by NOVA (*Buckanovich and Darnell, 1997*). The exon 16 cluster mutations did not cause any change in *Dcc* alternative splicing (*Figure 9B*). In contrast, when the intron 16 cluster was mutated, NOVA1 or NOVA2 could no longer promote *Dcc$_{long}$*, indicating that these sites are essential NOVA binding sites. The exon 17 cluster is in close proximity to the splice acceptor site for *Dcc$_{short}$*, and its mutation produced two additional RT-PCR products, generated from utilizing two downstream cryptic splice acceptors. However, NOVA1 or NOVA2 could still increase *Dcc$_{long}$* (*Figure 9B*). Overall, these results show that the intron 16 cluster is the main binding site for NOVA, and the binding promotes the production of *Dcc$_{long}$*.

The intron 16 cluster contains six YCAY repeats and is conserved in humans, with seven repeats present in human sequence. We next examined if the number of repeats could affect how effectively

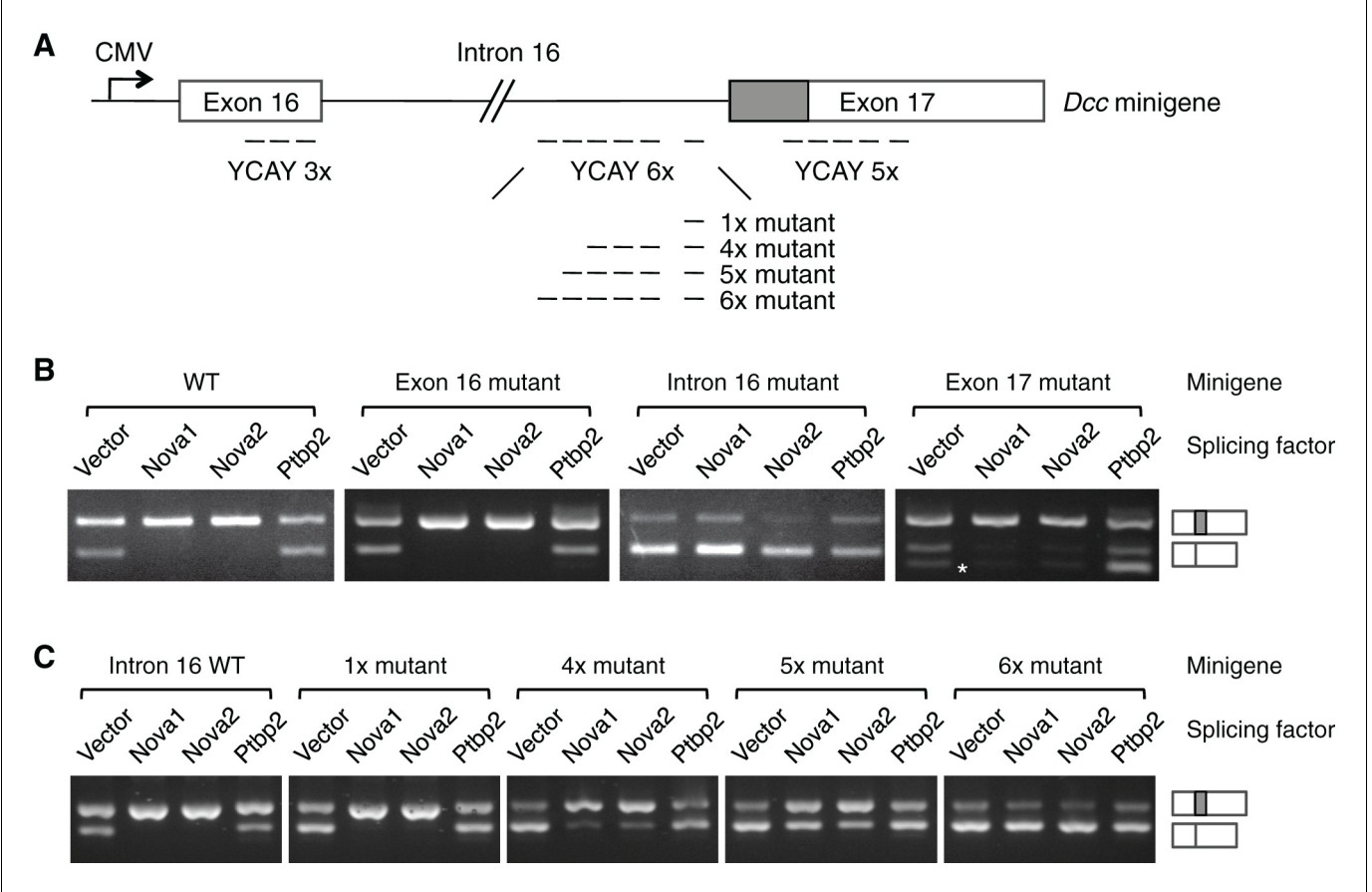

**Figure 9.** NOVA1/2 regulate *Dcc* alternative splicing. (**A**) Schematic of a *Dcc* minigene containing the genomic DNA between exons 16 and 17. The alternative sequence is shaded gray. Three candidate Nova binding sites, which are YCAY (Y=C/U) clusters, are located within exon 16, intron 16, and exon 17, respectively (dashed lines indicate the number of YCAY repeats). (**B**) Alternative splicing of wildtype and mutant minigenes. The two RT-PCR products correspond to $Dcc_{long}$ and $Dcc_{short}$, respectively. Two additional RT-PCR products are produced by exon 17 mutations (asterisk), from utilizing two downstream cryptic splice sites. Mutations in intron 16 block NOVA1/2 from increasing $Dcc_{long}$. (**C**) Alternative splicing of *Dcc* minigenes containing different numbers of mutations in intron 16. With an increasing number of mutations, NOVA1/2 gradually lose their ability to promote $Dcc_{long}$.

NOVA regulates *Dcc* alternative splicing by introducing mutations into a subset of the repeats. We found that mutations in just one repeat had little effect, but mutations in four repeats led to a partial reduction in $Dcc_{long}$ when *Nova1/2* were co-expressed (*Figure 9C*). With five and six mutated repeats, there were even further decreases in $Dcc_{long}$ and corresponding increases in $Dcc_{short}$ (*Figure 9C*). These results are consistent with the dose-dependent interaction between NOVA and their RNA targets (*Darnell, 2006*; *Figure 7B*), and further support that these sequences are bona fide NOVA binding sites.

## Discussion

Neuronal and axonal migration requires dynamic regulation of cell signaling, particularly evident as cells/axons encounter an intermediate target such as the midline of the central nervous system (*Dickson and Zou, 2010*; *Evans and Bashaw, 2010*; *Kaprielian et al., 2001*). Alternative splicing has been increasingly implicated as an important means to generate temporal and spatial specific functions for guidance molecules (*Chen et al., 2008*; *Park and Graveley, 2007*). To further understand the functional significance of alternative splicing, it is important to identify the relevant splicing factors and their targets. Through an in vivo RNAi screen against RBPs, we discovered that *Nova1/2* are key regulators of both neuronal and axonal migration in the spinal cord interneurons.

We found that *Nova* deficiency reduces the migration of mitotic progenitors and differentiated interneurons in the spinal cord (*Figure 3,4*). We observed similar defects in *Dcc* KOs, which were mostly uncharacterized before. Within the commissural interneurons, *Nova* dKO disrupts Netrin-induced axon outgrowth and ventral projection to the midline, also resembling the *Dcc* mutant (*Figure 5,6*). The phenotypic similarity between *Nova* and *Dcc* KOs strongly suggests that *Dcc* activity is affected by *Nova* dKO. Indeed, this is confirmed by the observations that *Dcc* alternative splicing is disrupted by *Nova* dKO (*Figure 7*), that *Nova* dKO defects can be largely rescued in vitro and in vivo by restoring *Dcc_{long}* (*Figure 8*, *Figure 8—figure supplement 1*, *Figure 1—source data 1*), and that NOVA controls *Dcc* alternative splicing in vitro (*Figure 9*). Also importantly, our results show that disrupting *Dcc* alternative splicing without affecting the total *Dcc* level leads to as severe defects as *Dcc* KO, underscoring the importance of alternative splicing for the gene function.

The defects in neuronal migration, axon outgrowth, and axon guidance in *Nova* and *Dcc* KOs appear to share a common feature, which is a temporal delay. The migration is slowed but not completely blocked (*Figure 4*). The axon outgrowth is initially defective but becomes normal after an extended culture period (*Figure 5*). The axon guidance defect also appears to be somewhat alleviated at E12.5 compared to earlier stages (*Figure 6*). This delay could be due to an incomplete loss of the gene function in the KOs. Alternatively, NOVA proteins have been shown to regulate many target genes including *Dcc* in a temporal specific manner, generating different ratios of splice variants at different developmental stages (*Yano et al., 2010*; also see accompanying paper by Saito et al.) As *Nova* deficiency changes the ratio between DCC variants, it may alter the developmental state of the neurons. Consequently, this leads to abnormal responsiveness of the neurons to the extracelluar environment and thus defects in neuronal and axonal migration.

The fact that the loss of *Dcc_{long}* in *Nova* dKOs cannot be compensated by the increase in *Dcc_{short}* demonstrates that the two isoforms are functionally distinct. The two DCC isoforms differ in the FN4-FN5 linker, with both FN4 and FN5 domains interacting with Netrin (*Xu et al., 2014*). A structure study of Netrin in complex with the DCC or NEO1 receptor found that DCC variants can bind to Netrin-1 with comparable affinities, but are likely to adopt distinct conformations in the ligand-receptor complex (*Xu et al., 2014*). Netrin and DCC_{short} form a continuous liand:receptor complex, whereas DCC_{long} is likely to form a 2:2 ligand:receptor complex with Netrin (*Xu et al., 2014*). Whether the presence of both isoforms and at different ratios can produce additional conformations of the ligand-receptor complex is completely unknown. How the architecturally distinct complexes can lead to different intracellular signaling also remains an intriguing question.

The defects seen in *Nova* dKOs could result from the loss in *Dcc_{long}*, the increase in *Dcc_{short}*, or the combination of both. We cannot yet distinguish between these possibilities. The rescue of *Nova* dKOs by DCC_{long}, but not by DCC_{short}, can also be interpreted in different ways. One possibility is that DCC_{long} is fully responsible for the gene function, whereas DCC_{short} has no activity. Another is that both isoforms are required and each has its specific activity. Only DCC_{long} can rescue *Nova* KOs because it is reduced in the mutants, while DCC_{short} is still present. Since the two *Dcc* isoforms are normally expressed at comparable levels in commissural axons at E11.5 (*Figure 7C*), it is likely that each isoform has its unique activity during neuronal migration and axon guidance.

*Dcc* alternative splicing was first identified using human neuroblastoma cells IMR32 (*Reale et al., 1994*). Compared with normal mouse brain tissues, IMR32 expresses a decreased level of *Dcc_{long}* and an elevated level of *Dcc_{short}* (*Reale et al., 1994*). These changes are in the same pattern as those in *Nova* dKOs. Thus, altered *Dcc* alternative splicing may also contribute to tumor development. Allelic loss of the 18q21 region encompassing *Dcc* is identified in about 70% of primary colorectal cancers and is also found in other types of cancers (*Mehlen and Fearon, 2004*). It remains to be seen if altered *Dcc* alternative splicing accounts for additional cases of colorectal cancers and other cancers. Furthermore, *Dcc* plays additional roles in the nervous system, such as in dendritic growth and guidance (*Furrer et al., 2003*; *Nagel et al., 2015*; *Suli et al., 2006*; *Teichmann and Shen, 2011*), and in synapse formation and function (*Colon-Ramos et al., 2007*; *Goldman et al., 2013*; *Horn et al., 2013*). Therefore, it is important to determine if *Dcc* alternative splicing is also important for other biological processes.

## Materials and methods

### Mice

*Nova1*, *Nova2*, and *Dcc* KOs were generated and described previously (*Fazeli et al., 1997*; *Huang et al., 2005*; *Jensen et al., 2000*; *Ruggiu et al., 2009*; accompanying paper by Saito et al.). All strains were outcrossed to the CD-1 strain. *Nova* double heterozygotes (dHet) were intercrossed to generate all nine genotypes used in the study.

### siRNAs

The siRNAs for candidate RNA-binding proteins were designed and synthesized by IDT (Coralville, IA). A pool of three siRNAs were used in the screen. Once phenotypes were seen, individual siRNAs were validated and the most potent siRNA was used for further phenotypic analyses. The sense sequences for the most potent *Nova* siRNAs are as follows: *Nova1* 5'tacaacctcagaccaccgttaatcctg3', *Nova2* 5'gaccatcgtgcagctccagaaggagac3', and pan-*Nova* 5'agccaccatcaagctgtctaagtccaa3'.

### cDNAs

*Dcc*, *Nova1*, *Nova2*, *Ptbp2* cDNAs were cloned from wildtype CD-1 mouse spinal cords. *Robo3$_{A.1}$*cDNA was generated previously (*Chen et al., 2008*). *Actb-gfp* (*egfp* in pCAGGS) and *Atoh1-gfp* markers were previously described (*Lumpkin et al., 2003*; *Matsuda and Cepko, 2004*).

### Whole embryo culture

WEC was carried out as previously described (*Chen et al., 2008*). For the RNAi screen, embryos were electroporated at E9.5 with siRNAs and *gfp* into one side of the spinal cord and were cultured for 40-48 hrs. The embryos were then fixed in 4% paraformaldehyde, cryopreserved in 30% sucrose, and embedded in OCT. 20 μm transverse sections were collected and examined using fluorescent microscopy. Alternatively, openbook preparation of the spinal cord was performed and examined by fluorescent microscopy as previously described (*Chen et al., 2008*).

### Quantification of phentoypes

For phenotypic quantification in all experiments, if *Nova* WT and dKO were not littermates, they were first normalized to the respective dHet littermates, and were then compared with each other. *Dcc* KOs were compared with WT littermate controls. To minimize developmental variation, we used embryos of comparable sizes and examined spinal cord tissues from the brachial level.

For quantification of cultured embryos labeled with *Actb-gfp*, neuroprogenitors in the VZ is represented as the ratio between the signal from the medial spinal cord and that from the lateral spinal cord. Axon ventral projection is represented as the ratio between the signal from axons at the ventral margin of the spinal and that from the beginning of the axon bundle. The signal intensity was measured using ImageJ (NIH, Bethesda, MD). For quantification of *Atoh1-gfp* labeled embryos, ventral axon projection is represented as the ratio between the distance from the dorsal margin of the spinal cord to the ventral most axons and the total height of the spinal cord. The distances were measured using ImageJ. In all phenotypic analyses, the defects are consistently seen in all embryos examined. The severity of defects is comparable between animals and between different sections of the same embryo. Representative images are shown in all figures. Numbers of animals and sections examined are listed in *Figure 1—source data 1*.

### Immunohistochemistry

IHC was carried out as previously descried (*Xu et al., 2014*). Antibodies used in the study include anti-PAX3/7 (PA1-107, Thermo Fisher, Waltham, MA, raised against PAX3 and cross reacts with PAX7), anti-BARHL2 (NBP2-32013, Novus Biologicals, Littleton, CO), anti-LHX5 (AF6290, R&D, Minneapolis, MN), anti-ISL1/2 (39.4D5, DSHB, Iowa City, IA), anti-ROBO3 (rabbit polyclonal, *Chen et al., 2008*), anti-TAG1 (4D7, DSHB), anti-pH3 (9701, CST, Danvers, MA), anti-Ki-67 (12202, CST), and anti-SOX2 (3728, CST).

## Dorsal spinal cord explant

DSC assay was performed as previously descried (*Xu et al., 2014*). For DSC taken from embryos grown in vivo, the explants were labeled with rhodamine-phalloidin (Thermo Fisher). The outgrowth was quantified as the ratio between the signal from the axons and that from the cell bodies, after background extraction. The signal intensity was measured using ImageJ. For the rescue experiments using the dorsal spinal cord explants, embryos were electroporated with cDNAs and *gfp* at E10.5 and cultured for 24 hrs. GFP-positive DSCs were microdissected and cultured. The length and total number of GFP positive axons were quantified using ImageJ. At least three embryos from each genotype and five explants from each embryo were quantified, with genotypes and treatments blinded.

## Quantitative and semi-quantitative RT-PCR

Spinal cord tissues were microdissected, and the dorsal and ventral halves were separated to distinguish the commissural and motor neuron populations. Total RNA was extracted using Trizol (Thermo Fisher), and reverse transcription was carried out using Maxima RT (Thermo Fisher). Quantitative PCR was performed using a Realplex$^2$ thermocycler (Eppendorf, Hamburg, Germany). Semi-quantitative PCR was performed to generate multiple isoforms in a single reaction and compare the relative expression by electrophoresis. The cycle number used in semi-quantitative PCR was determined by quantitative PCR to obtain products during the exponential amplification phase.

## In situ hybridization

ISH was performed as previously described (*Braissant and Wahli, 1998*). The antisense probes were in vitro transcribed using T7 polymerase and labeled with DIG (Digoxigenin, Roche, Basel, Switzerland). Sense probes were used as negative controls. For non-fluorescent ISH, AP (alkaline phosphatase)-conjugated anti-DIG antibody (Roche) was used to detect bound antisense probes, and was visualized using colorimetric AP substrates. For FISH, HRP (horse radish peroxidase)-conjugated anti-DIG antibody (Jackson ImmunoResearch, West Grove, PA) was used and the signal was visualized using the TSA (tyramide signal amplication) system (Perkin Elmer, Waltham, MA).

## Dorsal spinal cord neuron culture

To confirm exogenous protein expression, DSC neurons electroporated with *Dcc* or *Robo3* cDNA and *gfp* were microdissected, dissociated (0.05% trypsin, 0.5 mM EDTA), and cultured for 24 hrs in PDL (poly-D-Lysine, 100 µg/ml) coated culture dish in the culture medium (Neurobasal, 1x B27, 50 U/ml Pen/Strep, and 250 ng/ml Netrin-1). Cells were then fixed with 4% paraformaldehyde, and stained with anti-HA (3F10, Roche) and Alexa Fluor 594-conjugated secondary antibodies (Jackson ImmunoResearch).

## Splicing assay

*Dcc* genomic DNA that spans exons 16 and 17 (5.6 kb total) was PCR amplified from mouse spinal cords and cloned into the pDEST26 gateway vector containing a CMV promoter (Thermo Fisher). *Dcc* minigene was transfected into HEK293T cells together with the splicing factors or an empty vector at a 1:1 ratio. Cells were cultured for 48 hr and the total RNA was collected using Trizol (Thermo Fisher). Reverse transcription was carried out from a T7 promoter (present in pDEST26) using SMARTScribe reverse transcriptase (Clontech, Mountain View, CA), and semi-quantitative PCR was performed to amplify multiple isoforms. Point mutations were introduced by PCR reactions using Pfu polymerase (Agilent, Santa Clara, CA), and were confirmed by DNA sequencing. A V5 tag at the C-terminus of NOVA1, NOVA2, and PTBP2 was used to confirm protein expression using western blotting.

**Primers used for quantitative PCR**

| Gene | Amplicon | Forward primer | Reverse primer |
|------|----------|----------------|----------------|
| *Dcc* | $Dcc_{long}$ | tctcattatgtaatctccttaaaagc | gggaaatcatcaagcaaaggataataa |
| | $Dcc_{short}$ | tctcattatgtaatctccttaaaagc | ggtggagacatctgttatggaacga |
| | Total *Dcc* (common region) | tctcattatgtaatctccttaaaagc | ctgttatggaacgagtggtggc |
| *Neo1* | $Neo1_{long}$ | tgttattaatgctccatacactccag | ccaggtaatccttatggtgtcgt |

| | | | |
|---|---|---|---|
| | Neo1$_{short}$ | cagacctcacacagtgccagatccc | ccaggtaatccttatggtgtcgt |
| | Total Neo1 (common region) | tgccagatcccactcccat | ccaggtaatccttatggtgtcgt |
| Robo3 | Robo3$_A$ | tggaggggcttacggctccc | tagcgcagcatagcgcagcc |
| | Robo3$_B$ | acctggtcttcccccagttgct | gctcgcccctggaaaccacc |
| | Robo3$_{.1}$ | ccaccacccttgccaccacc | ccaggcctcttccgcagcac |
| | Robo3$_{.2}$ | ccaccacccttgccaccacc | gcaagcctccagtcccctccc |
| | Robo3 intron 19 | cctagtccctgcccctgacca | gagggactccgaggtgggtgg |
| | Robo3 intron 20 | ttggccctgctgcctacccat | tgccccaggaagctgacgga |
| **Primers used for semi-quantitative PCR (amplifying multiple isoforms)** | | | |
| Gene | Amplicon | Forward primer | Reverse primer |
| Dcc | Dcc common | tctcattatgtaatctccttaaaagc | tcacagcctcatgggtaagag |
| Neo1 | Neo1 common | ttggcgaaggcatcccccc | ccaggtaatccttatggtgtcgt |

## Acknowledgements

This work was supported by grants: Boettcher Foundation (to ZC); National Institutes of Health (NIH) R01EY024261 (to HJJ); Japan Society for the Promotion of Science (JSPS) postdoctoral fellowship for research abroad (to YS); NIH R01 NS34389, NS069473, NS081706 and Howard Hughes Medical Institute (HHMI) (to RBD). We thank Ben Weaver and Paul Muhlrad for critical comments on the manuscript.

## Additional information

### Funding

| Funder | Grant reference number | Author |
|---|---|---|
| Japan Society for the Promotion of Science | Postdoctoral Fellowship for Research Abroad | Yuhki Saito |
| National Institutes of Health | NIH R01 NS34389 | Robert B Darnell |
| National Institutes of Health | NS069473 | Robert B Darnell |
| National Institutes of Health | NS081706 | Robert B Darnell |
| Howard Hughes Medical Institute | | Robert B Darnell |
| National Institutes of Health | R01EY024261 | Harald J Junge |
| Boettcher Foundation | New Investigator Award | Zhe Chen |
| Boettcher Foundation | OCG5499B | Zhe Chen |

The funders had no role in study design, data collection and interpretation, or the decision to submit the work for publication.

### Author contributions

JCL, Acquisition of data; YS, Drafting or revising the article; RBD, Drafting or revising the article, Contributed unpublished essential data or reagents; MT-L, Conception and design, Drafting or revising the article; HJJ, Analysis and interpretation of data, Drafting or revising the article, Contributed unpublished essential data or reagents; ZC, Conception and design, Acquisition of data, Analysis and interpretation of data, Drafting or revising the article

### Author ORCIDs

Robert B Darnell, http://orcid.org/0000-0002-5134-8088
Zhe Chen, http://orcid.org/0000-0003-0683-9491

### Ethics

Animal experimentation: This study was performed in strict accordance with the recommendations in the Guide for the Care and Use of Laboratory Animals of the National Institutes of Health. All of the animals were handled according to the approved institutional animal care and use committee (IACUC) protocol (#1310.02) of the University of Colorado at Boulder.

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
