## [Decision Letter]

Thank you for submitting your article "NOVA RNA-binding proteins regulate *Dcc* alternative splicing during neuronal migration and axon guidance" for consideration by *eLife*. Your manuscript has now been evaluated by three reviewers, one of whom is a member of the Board of Reviewing Editors and it has also been evaluated by a Senior Editor.

The reviewers have discussed the reviews with one another and the Reviewing Editor has drafted this decision to help you prepare a revised submission.

As you will see from the reviews, the reviewers found the paper to be interesting but identified some weaknesses that you must address before we can consider publication of the work. Some of the issues require better data analyses while others require revision of the text to consider additional interpretations. We also suggest that you read the comments on the accompanying paper from the Darnell Lab to consider some of text revisions. The separate reviews are also included at the end of this summary:

There is a main weakness that we hope you can address: If you can enhance the characterization of the anatomical phenotypes, we believe the manuscript would benefit tremendously. Currently you use fairly coarse read-outs to characterize the anatomical phenotypes. Many assessments are semi-quantitative and do not allow for an estimate of what fraction of the manipulated cells are affected. There is no doubt that the phenotypes exist and are rescued by the overexpression of the long DCC splice isoform but it remains unknown how severe the phenotypes are and whether they persist at later stages of development. Please attempt to provide more quantitative analyses and report if the phenotypes persist at later stages.

The following are additional comments that should be addressed by editing the text.

1) The abstract of the manuscript focuses almost entirely on alternative splicing yet this aspect of the work does not begin to be addressed until late in the manuscript – Figure 7 and subsection “*Nova* dKO disrupts *Dcc* alternative splicing”. The authors should consider how to balance better the content of the abstract to reflect the full body of the manuscript. Perhaps some of the data, particularly the cell migration results, can be reduced.

2) *Nova* KO delays cell migration in the dorsal spinal cord, as does the DCC KO. It seems possible that this delay in cell migration may underlie the later-arising axon defect. For example, if axons grow out later in development they would appear to be shorter in vivo (as seen in Figure 1, and Figure 1—figure supplement 2, and Figure 2) and potentially defective due to growing in an abnormally mature spinal cord. A delay in axon outgrowth could also provide a simple explanation for the results in Figure 5 where the *Nova* 1/2 dKO shows no neurite outgrowth at 24 hr defect. The authors should address this point in the manuscript.

3) ROBO3 was examined for changes in alternative splicing in the *Nova* 1/2 KO mutant and no differences were found. But is ROBO3 a known target of *Nova*? It would be informative to include data from another receptor that is a known *Nova* target, other than DCC, to investigate here.

4) The data can be interpreted in several different ways. The authors conclude that DCC long is required for ventral migration. An alternative view, would be that DCC_short_ antagonizes ventral migration. By inhibiting splicing the ratio of DCC_short_ to DCC_long_ increases. The data do not rule out the possibility that migration phenotype could reflect reduction in DCC_long_, an increase in DCC_short_ or both. Furthermore, normal ventral migration may require both. That is DCC_long_ and DCC_short_ may form a heterodimer and this heterodimer, and a certain level of it, may be essential for normal migration. The discussion in the other manuscript that considers the relationship between different forms of DCC and its function should be incorporated into this manuscript.

Reviewer #1:

The manuscript investigates the function of the RNA-binding protein, *Nova*, in neuronal development in the spinal cord and reports that loss of function leads to defects in cell migration and axon guidance. The commissural neurons in the dorsal spinal cord exhibit defects in midline crossing similar to that described previously for DCC knockouts. Evidence is presented to show that *Nova* controls the splicing of DCC isoforms and that when *Nova* is knocked out, the long isoform is decreased. Expression of the long, but not the short, DCC isoform rescues the *Nova*-/- cell migration defect and the axon phenotype in vivo and in vitro. The authors conclude that splicing of DCC is important for its function in commissural neurons and that *Nova* controls this alternative splicing.

This is an interesting study that provides quite robust evidence showing that *Nova* proteins control splicing of DCC and demonstrates a key role of the long DCC isoform in controlling cell migration in the dorsal spinal cord and axon growth to the midline. Overall, the study has been well executed and the results are mostly convincing. It is perhaps surprising that alternative splicing of just one target, DCC, is identified as the key mediator of the phenotype given that *Nova* has been shown previously to have many target mRNAs. However, the evidence showing the rescue of the migration and axon phenotypes with DCC_long_ but not DCC_short_ is quite persuasive. [Minor comments not shown.]

Reviewer #2:

The authors report cell migration and axon guidance phenotypes of spinal dorsal interneurons in rodent loss of function models. They conclude that the RNA binding proteins *Nova* 1 and 2 contribute to the alternative splicing regulation of the cell surface receptor DCC. Loss of *Nova1/2* or DCC result in altered migration and axon trajectory patterns of commissural interneurons. These defects can be rescued by overexpression of one DCC alternative splice variant but not by overexpression of a DCC alternative splice variant that persists in the *Nova1/2* KO embryo.

Overall, the work is interesting as it links a specific alternative splicing event in an axon guidance receptor, an RNA binding protein and potential phenotypes in a fairly intact system (embryo culture). The two main weaknesses of the study are:

1) The authors use fairly coarse read-outs to characterize the anatomical phenotypes. Many assessments are semi-quantitative and do not allow for an estimate of what fraction of the manipulated cells are affected. There is no doubt that the phenotypes exist and are rescued by the overexpression of the long DCC splice isoform but it remains unknown how severe the phenotypes are and whether they persist at later stages of development.

2) The other major (and probably more severe) weakness is the lack of mechanistic insight. The authors attempt to test whether the long and short DCC splice variants might differ in stability, expression, or localization and do not observe changes for ectopically expressed forms. However, it remains unclear whether the long variant indeed engages in some specific protein complexes at the cell surface. I should be noted that the authors openly discuss this issue in the manuscript and I feel that the work is an important contribution, even if the mechanism remains unclear.

[Minor comments not shown.]

Reviewer #3:

In this paper, Leggere and colleagues explore the role of NOVA proteins, splicing factors highly enriched in the CNS, in regulating development of commissural neurons in the mouse spinal cord. Commissural neurons are born in the dorsal spinal cord, migrate laterally, and then ventrally to populate regions in the dorsal half of the spinal cord. These neurons send axons ventrally across the floor plate to the contralateral side of the spinal cord. Removal of both NOVA1 and NOVA2, first shown using RNAi and then through traditional mutations, disrupts the development of these neurons. Data are presented that these neurons differentiate normally, as assessed by various markers, but they show abnormalities in cell migration (both laterally and ventrally) and in axon guidance to the ventral midline. As these phenotypes are similar to DCC loss-of-function mutations, the authors sought to explore the relationship between NOVA1/2 and DCC function. DCC is a receptor for netrin; netrin is expressed in the ventral half of the spinal cord and plays an important role in regulating the ventral migration of commissural neurons and the guidance of the growth cones to the floor plate.

The authors demonstrate that NOVA1/2 regulates alternative splicing of DCC. In the DCC pre-mRNA there are two alternative acceptor sites downstream of exon 16. The use of 5'-site generates a protein with an insert of 20AA longer than the protein in the downstream site. These are referred to as DCC_long_ and DCC_short_, respectively. The authors demonstrate that the NOVA1/2 phenotype can be rescued by expression of DCC_long_ but not DCC_short_. Using a cell-culture based splicing assay, they showed that NOVA1/2 regulates this splice and does so through specific interactions with YCAY sites (canonical NOVA1/2 recognition sequences) within the intron separating exon 15 and 16.

The paper is clearly written and the data in the figures are easy to follow. The data can be interpreted in several different ways. The authors conclude that DCC long is required for ventral migration. An alternative view, would be that DCC_short_ antagonizes ventral migration. By inhibiting splicing the ratio of DCC_short_ to DCC_long_ increases. The data do not rule out the possibility that migration phenotype could reflect reduction in DCC_long_, an increase in DCC_short_ or both. Furthermore, normal ventral migration may require both. That is DCC_long_ and DCC_short_ may form a heterodimer and this heterodimer, and a certain level of it, may be essential for normal migration. The discussion in the other manuscript that considers the relationship between different forms of DCC and its function should be incorporated into this manuscript.

The paper is appropriate for *eLife*.

---

## [Author Response]

There is a main weakness that we hope you can address: If you can enhance the characterization of the anatomical phenotypes, we believe the manuscript would benefit tremendously. Currently you use fairly coarse read-outs to characterize the anatomical phenotypes. Many assessments are semi-quantitative and do not allow for an estimate of what fraction of the manipulated cells are affected. There is no doubt that the phenotypes exist and are rescued by the overexpression of the long DCC splice isoform but it remains unknown how severe the phenotypes are and whether they persist at later stages of development. Please attempt to provide more quantitative analyses and report if the phenotypes persist at later stages.

We think this comment refers to the quantification of the cultured embryos. The phenotypic analyses and quantifications of knockout embryos and of neuronal explants were carried out using previously published methods. It is technically challenging to assess the phenotypes in cultured embryos labeled with *Actb-gfp (βactin-gfp*). The GFP is expressed throughout the cell bodies and the neurites and the neurons are closely positioned, making it difficult to discern individual neurons. We thus compared the overall signal from the ventricular zone with that from the lateral spinal cord to assess neuronal migration. It is also difficult to count GFP+ axons as they form bundles, and to measure the axon length as the cell bodies are positioned at different dorsoventral level. With that said, we have performed the following quantifications to improve our data.

1) We have now quantified the ventral projection of the commissural axons using embryos labeled with the Atoh1-gfp (Math1-gfp) marker. Atoh1 is expressed by neurons born at the dorsal margin of the spinal cord, whose axons extend across the whole spinal cord to the ventral midline. We measured the distance from the dorsal margin of the spinal cord to the ventral most GFP+ axons and the total height of the spinal cord. The ratio between the two represents the portion of the spinal cord that the leading axons have traversed (Figure 1—figure supplement 2, Figure 8—figure supplement 1). Consistent with the quantification using Actb-gfp, both Nova and Dcc KOs display a significant decrease in the ventral axon projection toward the midline. And Dcc_long_ is able to restore the ventral projection in Nova KOs (quantification now included in Figure 1—figure supplement 2, Figure 8—figure supplement 1, and [Supplementary-material SD1-data]).

2) We have also included the number of embryos and sections analyzed and the fraction of sections where we observed the phenotypes in neuronal migration and axon guidance ([Supplementary-material SD1-data]). We would like to emphasize that in all phenotypic analyses, the defects are consistently seen in all embryos examined. The severity of defects is comparable between animals and between different sections of the same embryo. Representative images are shown in all figures.

To address if the phenotypes persist at later stages, we have performed the following experiments.

1) We were able to culture some mouse embryos until the equivalent stage of E12 (half a day older than those shown in Figure 1 and Figure 8). It is technically difficult to grow the embryos any longer due to reduced survival. In the older embryos, the cell migration defect appeared to be slightly improved in *Nova* KOs, but there are still fewer axons than in controls . We did not include the data in the manuscript as we feel that the data using the KO embryos is more quantitative (see below).

2) Using knockout embryos, we examined axon guidance at a later stage and the data is incorporated into Figure 6 (ROBO3 staining at E12.5). We found that at E12.5, the axon guidance defect in both *Nova* and *Dcc* KOs is still present, but is somewhat weaker than at earlier stages. Examination of even later stages is difficult, as the ROBO3 and TAG-1 markers become downregulated after E12.5 and E11.5, respectively. We have also provided further discussion on the temporal profile of the defects in the manuscript.

Author response image 1.**DOI:**
http://dx.doi.org/10.7554/eLife.14264.025

In a previous study where we used the embryo culture to determine the function of different Robo3 receptor isoforms (Chen et al., Neuron, 2008), we assessed what fraction of the manipulated cells is affected. We found that the exogenously expressed ROBO3_A.1_ protein mostly overlaps with the GFP marker and ROBO3_A.1_ expression is able to rescue the midline crossing defect in Robo3 KOs to a large extent. We estimated that around 80% of Robo3 KO axons that express Robo3_A.1_ are able to cross the midline, whereas no axon expressing GFP alone can cross.

Author response image 2.**DOI:**
http://dx.doi.org/10.7554/eLife.14264.026

Using the same approach, we attempted to assess what fraction of neurons that exogenously express DCC is affected in this study. However, due to the presence of endogenous DCC in *Nova* KOs, we were unable to specifically detect the exogenous protein using anti-DCC. The exogenous DCC is tagged with HA at the C-terminus. Although we could detect the HA tag in dissociated neurons (Figure 8), we were unable to detect any signal in tissue sections. However, given that the rescue of *Nova* dKO by *Dcc*_long_ is observed in all animals and in all sections examined, and that the rescue is to a large degree (Figure 8, Figure 8—figure supplement 1, [Supplementary-material SD1-data]), we think most manipulated cells are affected, similar to what we have observed before in the Robo3 study.

The following are additional comments that should be addressed by editing the text.

1) The abstract of the manuscript focuses almost entirely on alternative splicing yet this aspect of the work does not begin to be addressed until late in the manuscript – Figure 7 and subsection “Nova dKO disrupts Dcc alternative splicing”. The authors should consider how to balance better the content of the abstract to reflect the full body of the manuscript. Perhaps some of the data, particularly the cell migration results, can be reduced.

We have modified the abstract to better reflect the overall findings of the study. We have also reduced the text describing the cell migration results.

2) Nova KO delays cell migration in the dorsal spinal cord, as does the DCC KO. It seems possible that this delay in cell migration may underlie the later-arising axon defect. For example, if axons grow out later in development they would appear to be shorter in vivo (as seen in Figure 1, and Figure 1—figure supplement 2, and Figure 2) and potentially defective due to growing in an abnormally mature spinal cord. A delay in axon outgrowth could also provide a simple explanation for the results in Figure 5 where the Nova 1/2 dKO shows no neurite outgrowth at 24 hr defect. The authors should address this point in the manuscript.

We agree that the cell migration and axon outgrowth/guidance defects seen in *Nova* and *Dcc* KOs may share a common underlying mechanism, which is a temporal delay. Per reviewer 2’ suggestion, we have now included axon guidance data from a later stage (Figure 6, ROBO3 staining at E12.5). We found that at E12.5, the axon guidance defect in both *Nova* and *Dcc* KOs is still present, but is somewhat weaker than at earlier stages. Together, these data suggest that a temporal delay may underlie the defects in both cell migration and axon guidance. We have now included further discussion to address this point.

3) ROBO3 was examined for changes in alternative splicing in the Nova 1/2 KO mutant and no differences were found. But is ROBO3 a known target of Nova? It would be informative to include data from another receptor that is a known Nova target, other than DCC, to investigate here.

Robo3 has not been shown to be a NOVA target. A previous transcriptome-wide HITS-CLIP from the Darnell group did not identify any candidate NOVA binding sites within Robo3 (Zhang et al., Science, 2010). However, due to the low expression level of Robo3 in the examined tissue (P0 brain), the result does not rule out Robo3 as a potential target (personal communication with R.B. Darnell). We examined Robo3 as it functions as a coreceptor with *Dcc* and is known to undergo alternative splicing.

Besides *Dcc* and Robo3, we also examined *Neo1*, which is a *Dcc* homolog and a known NOVA target. We have edited the text to make this point clear. We found that its alternative splicing at exons 18 and 27 was not significantly altered by *Nova* KO in the dorsal spinal cord (Figure 7—figure supplement 3).

To further address the reviewer’s question, we have now examined the *Epha5* receptor (Figure 7—figure supplement 3). *Epha5* is a known NOVA target but has not been shown to play a role in the development of the spinal interneurons. We found no change in *Epha5* alternative splicing at exon 7 in the dorsal spinal cord.

Interestingly, the accompanying paper by Saito et al. shows that the alternative splicing of *Neo1* exon 27 and *Epha5* exon 7 is significantly altered in E18.5 *Nova2* KO cortex. The distinct observations between E11.5 dorsal spinal cord and E18.5 cortex show that NOVA functions in a temporospatial specific manner.

4) The data can be interpreted in several different ways. The authors conclude that DCC_long_ is required for ventral migration. An alternative view, would be that DCC_short_ antagonizes ventral migration. By inhibiting splicing the ratio of DCC_short_ to DCC_long_ increases. The data do not rule out the possibility that migration phenotype could reflect reduction in DCC_long_, an increase in DCC_short_ or both. Furthermore, normal ventral migration may require both. That is DCC_long_ and DCC_short_ may form a heterodimer and this heterodimer, and a certain level of it, may be essential for normal migration. The discussion in the other manuscript that considers the relationship between different forms of DCC and its function should be incorporated into this manuscript.

We fully agree that there are several alternative models on DCC isoform functions. Our data does not yet allow us to distinguish between the different possibilities outlined by the reviewer. We have now included more thorough discussion on the possible interpretations of our results.

Reviewer #2:

The authors report cell migration and axon guidance phenotypes of spinal dorsal interneurons in rodent loss of function models. They conclude that the RNA binding proteins Nova 1 and 2 contribute to the alternative splicing regulation of the cell surface receptor DCC. Loss of Nova1/2 or DCC result in altered migration and axon trajectory patterns of commissural interneurons. These defects can be rescued by overexpression of one DCC alternative splice variant but not by overexpression of a DCC alternative splice variant that persists in the Nova1/2 KO embryo.

Overall, the work is interesting as it links a specific alternative splicing event in an axon guidance receptor, an RNA binding protein and potential phenotypes in a fairly intact system (embryo culture). The two main weaknesses of the study are:

1) The authors use fairly coarse read-outs to characterize the anatomical phenotypes. Many assessments are semi-quantitative and do not allow for an estimate of what fraction of the manipulated cells are affected. There is no doubt that the phenotypes exist and are rescued by the overexpression of the long DCC splice isoform but it remains unknown how severe the phenotypes are and whether they persist at later stages of development.

We think this comment refers to the quantification of the cultured embryos. The phenotypic analyses and quantifications of knockout embryos and of neuronal explants were carried out using previously published methods. It is technically challenging to assess the phenotypes in cultured embryos labeled with *Actb-gfp (βactin-gfp*). The GFP is expressed throughout the cell bodies and the neurites and the neurons are closely positioned, making it difficult to discern individual neurons. We thus compared the overall signal from the ventricular zone with that from the lateral spinal cord to assess neuronal migration. It is also difficult to count GFP+ axons as they form bundles, and to measure the axon length as the cell bodies are positioned at different dorsoventral level. With that said, we have performed the following quantifications to improve our data.

1) We have now quantified the ventral projection of the commissural axons using embryos labeled with the *Atoh1-gfp (Math1-gfp*) marker. *Atoh1* is expressed by neurons born at the dorsal margin of the spinal cord, whose axons extend across the whole spinal cord to the ventral midline. We measured the distance from the dorsal margin of the spinal cord to the ventral most GFP+ axons and the total height of the spinal cord. The ratio between the two represents the portion of the spinal cord that the leading axons have traversed (Figure 1—figure supplement 2, Figure 8—figure supplement 1). Consistent with the quantification using *Actb-gfp*, both *Nova* and *Dcc* KOs display a significant decrease in the ventral axon projection toward the midline. And *Dcc*_long_ is able to restore the ventral projection in *Nova* KOs (quantification now included in Figure 1—figure supplement 2, Figure 8—figure supplement 1, and [Supplementary-material SD1-data]).

2) We have also included the number of embryos and sections analyzed and the fraction of sections where we observed the phenotypes in neuronal migration and axon guidance ([Supplementary-material SD1-data]). We would like to emphasize that in all phenotypic analyses, the defects are consistently seen in all embryos examined. The severity of defects is comparable between animals and between different sections of the same embryo. Representative images are shown in all figures.

To address if the phenotypes persist at later stages, we have performed the following experiments.

1) We were able to culture some mouse embryos until the equivalent stage of E12 (half a day older than those shown in Figure 1 and Figure 8). It is technically difficult to grow the embryos any longer due to reduced survival. In the older embryos, the cell migration defect appeared to be slightly improved in *Nova* KOs, but there are still fewer axons than in controls. We did not include the data in the manuscript as we feel that the data using the KO embryos is more quantitative (see below).

2) Using knockout embryos, we examined axon guidance at a later stage and the data is incorporated into Figure 6 (ROBO3 staining at E12.5). We found that at E12.5, the axon guidance defect in both *Nova* and *Dcc* KOs is still present, but is somewhat weaker than at earlier stages. Examination of even later stages is difficult, as the ROBO3 and TAG-1 markers become downregulated after E12.5 and E11.5, respectively. We have also provided further discussion on the temporal profile of the defects in the manuscript.

In a previous study where we used the embryo culture to determine the function of different Robo3 receptor isoforms (Chen et al., Neuron, 2008), we assessed what fraction of the manipulated cells is affected. We found that the exogenously expressed ROBO3_A.1_ protein mostly overlaps with the GFP marker and ROBO3_A.1_ expression is able to rescue the midline crossing defect in Robo3 KOs to a large extent. We estimated that around 80% of Robo3 KO axons that express Robo3_A.1_ are able to cross the midline, whereas no axon expressing GFP alone can cross.

Using the same approach, we attempted to assess what fraction of neurons that exogenously express DCC is affected in this study. However, due to the presence of endogenous DCC in *Nova* KOs, we were unable to specifically detect the exogenous protein using anti-DCC. The exogenous DCC is tagged with HA at the C-terminus. Although we could detect the HA tag in dissociated neurons (Figure 8), we were unable to detect any signal in tissue sections. However, given that the rescue of *Nova* dKO by *Dcc*_long_ is observed in all animals and in all sections examined, and that the rescue is to a large degree (Figure 8, Figure 8—figure supplement 1, [Supplementary-material SD1-data]), we think most manipulated cells are affected, similar to what we have observed before in the Robo3 study.

2) The other major (and probably more severe) weakness is the lack of mechanistic insight. The authors attempt to test whether the long and short DCC splice variants might differ in stability, expression, or localization and do not observe changes for ectopically expressed forms. However, it remains unclear whether the long variant indeed engages in some specific protein complexes at the cell surface. I should be noted that the authors openly discuss this issue in the manuscript and I feel that the work is an important contribution, even if the mechanism remains unclear.

We agree that our study does not yet address the mechanism underlying the distinct activities of DCC isoforms. There are many possibilities to be examined. This study revealed the functional importance of *Dcc* alternative splicing and identified the relevant splicing factor, which would be the first step in determining how each DCC variant contributes to the gene function during cell migration and axon guidance.

Reviewer #3:

[…] The paper is clearly written and the data in the figures are easy to follow. The data can be interpreted in several different ways. The authors conclude that DCC long is required for ventral migration. An alternative view, would be that DCC_short_ antagonizes ventral migration. By inhibiting splicing the ratio of DCC_short_ to DCC_long_ increases. The data do not rule out the possibility that migration phenotype could reflect reduction in DCC_long_, an increase in DCC_short_ or both. Furthermore, normal ventral migration may require both. That is DCC_long_ and DCC_short_ may form a heterodimer and this heterodimer, and a certain level of it, may be essential for normal migration. The discussion in the other manuscript that considers the relationship between different forms of DCC and its function should be incorporated into this manuscript.

The paper is appropriate for eLife.

We fully agree that there are several alternative models on DCC isoform functions. Our data does not yet allow us to distinguish between the different possibilities outlined by the reviewer. We have now included more thorough discussion on the possible interpretations of our results.